# Adaptive optical waveguide system for large-area and overheating-preventing phototherapy in deep tissue

Zhenhao Wang [1], Zhaoxiang Yang[1], Yingchao Ma[1], Shanzhi Lyu[1], Xiaohua Hu[2] & Yapei Wang [1] ✉

Phototherapy, valued for its non-toxicity, selectivity, and minimal trauma, is predominantly applied to treat superficial diseases due to the limited penetration of light through tissues. While optical-fiber-assisted interventional phototherapy addresses this limitation, it lacks an immediate mechanism to mitigate overheating of surrounding healthy tissues. To improve the biosafety of interventional phototherapy, we develop an adaptive optical waveguide system (AOWS) based on a negative feedback modulation mechanism. The AOWS employs a thermally responsive liquid as the waveguide gating material, characterized by a precisely tunable low critical solution temperature (LCST). When the temperature surpasses the LCST, one-dimensional light propagation is scattered, providing effective thermal regulation. Furthermore, the design supports adjustable optical fiber outlets, with convex or concave configurations and varied curvatures, enabling precise control of the divergence angle. Superior to conventional optical fibers with smaller divergence angles, the AOWS facilitates closer placement to the lesion site, delivering a larger illumination area while significantly reducing the light pathway through normal tissue. More importantly, it provides thermal protection almost like an "on-off" switch without relying on irradiation power, ensuring enhanced safety and efficacy.

Phototherapy, referring to the treatment of abnormal tissues by light irradiation, appears to be an evolving subject caring about light relevant materials and technologies[1]. In principle, photons induce the generation of hyperthermia effect or reactive oxygen species as a result of photophysical or photochemical processes once light interacts with the photoactive substances[2–6]. Both processes subsequently lead to cell apoptosis within the lesion tissues[7–12]. However, light can hardly enter the deeper sites of interest through the skin tissue, where light is mostly scattered or absorbed by biological components and converted into thermal heat[13,14]. Following the way of shedding light on the skin (Fig. 1a), only the superficial lesions can receive effective phototherapy; otherwise, the normal tissue along the light pathway

might be thermally injured in order to deepen the light penetration by increasing the light intensity[15,16].

Deeper phototherapy can be achieved when light is delivered directly to the lesion through optical fibers[13,14,17–29]. Such an optic fiber-assisted phototherapy, referred to as interventional phototherapy, encompasses two primary approaches: placing the optical fiber close but noncontact to the lesion tissue (Fig. 1b) and directly inserting the optical fiber into the lesion tissue[20–29] (Fig. 1c). Theoretically, the noncontact approach provides a larger light exposure area, making it suitable for lesions with considerable size, whereas the direct insertion method is more appropriate for smaller lesions. In the noncontact approach (Fig. 1b), the light coverage area is determined by the

[1]School of Chemistry and Life Resources, Renmin University of China, Beijing, China. [2]Department of Burns and Plastic Surgery, Beijing Jishuitan Hospital, Capital Medical University, Beijing, China. ✉e-mail: yapeiwang@ruc.edu.cn

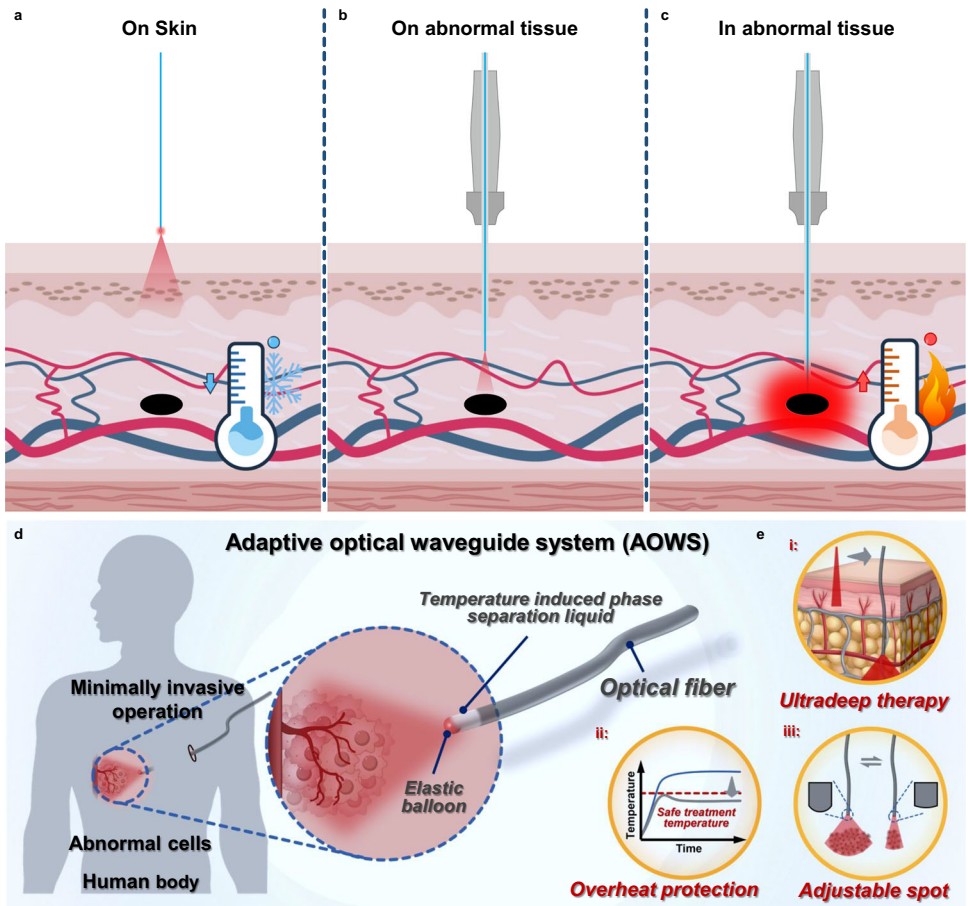

**Fig. 1 | The overall design of Adaptive Optical Waveguide system (AOWS). a–c** Schematic illustration of the three different light irradiation modes. **d** Principles of AOWS. **e** Advantages of AOWS relative to other light-delivering systems.

divergence angle of the optical fiber, which is positively correlated with the relative refractive index difference between its core and cladding (Eq. (1) in the "Method" section). For commercially available optical fibers, divergence angles are typically less than ±10° in a water medium. Consequently, the optical fiber tip must be positioned approximately 28 mm from the lesion to ensure complete light coverage for a lesion with a diameter of 10 mm[29]. Supposing that the tissue situated between the optical fiber tip and the lesion is normal tissue, the light energy would be attenuated significantly before it reaches the lesion area owing to the light absorption and scattering by normal tissues. As a result, the normal tissue absorbs light energy and gradually heats up, potentially leading to thermal injury over time. For the way of inserting the optical fiber directly into the lesion[24–29] (Fig. 1c), achieving a sufficiently high temperature at the central tissue is crucial to ensure thorough diffusion of sufficient thermal energy throughout the entire abnormal tissue, along with the tissue carbonization at the optical fiber outlet (further detailed in the "Method" section and Supplementary Figs. S1–S3). It should also be noted that the current interventional phototherapy using optical fiber technologies is unable to sense and respond immediately to the changes in subcutaneous temperature. It still remains a challenge to prevent the normal tissues from overheating during the phototherapy process.

In recent years, photothermal therapy (PTT) has progressed toward precise temperature monitoring and autonomous thermal control[26–28,30–32]. For accurate temperature monitoring, Wei et al. adjusted irradiation power density and exposure duration, employing repeated irradiation cycles to maintain therapeutic efficacy while avoiding thermal injury[30]. Zheng et al. developed a photothermo-electric cobalt-infused chitosan nanocomposite hydrogel that exhibits

a linear correlation between temperature shifts and resistance changes, enabling precise subcutaneous temperature measurements to prevent overheating[32]. However, these approaches cannot respond instantaneously to thermal excursions and instead rely on manual adjustments of laser power based on real-time temperature feedback. In the area of autonomous temperature regulation, Zhu et al. proposed a photomedicine strategy using a temperature-adaptive hydrogel optical fiber waveguide[26–28]. This strategy harnesses the phase transition of a responsive material as an optical switch to limit the core temperature of the photothermal agent. Because the fiber tip and the photothermal medium are in intimate contact, thermal equilibrium is reached rapidly. In practice, however, variations in tissue thermal conductivity and laser power produce substantial differences in the resulting steady state thermal distribution and ablation volume. Ensuring that the treated region precisely matches the lesion geometry, therefore, requires tuning both the phase transition temperature and the laser output with a high degree of precision. Without such precision, neighboring healthy tissues remain vulnerable to unintended thermal damage, a challenge that is difficult to surmount in clinical applications.

Herein, we developed an adaptive optical waveguide system (AOWS) for deeper phototherapy with tissue protection based on a negative feedback modulation mechanism (Fig. 1d). Unlike traditional optical fiber technologies, the AOWS possesses the high flexibility to adapt to soft tissues, the capability of light gating in response to temperature change, and a fiber outlet with tunable divergence angles. Besides the handiwork of the optical waveguide system, we presented an ionic liquid whose low critical solution temperature (LCST) in aqueous solution could be precisely tuned by either changing the

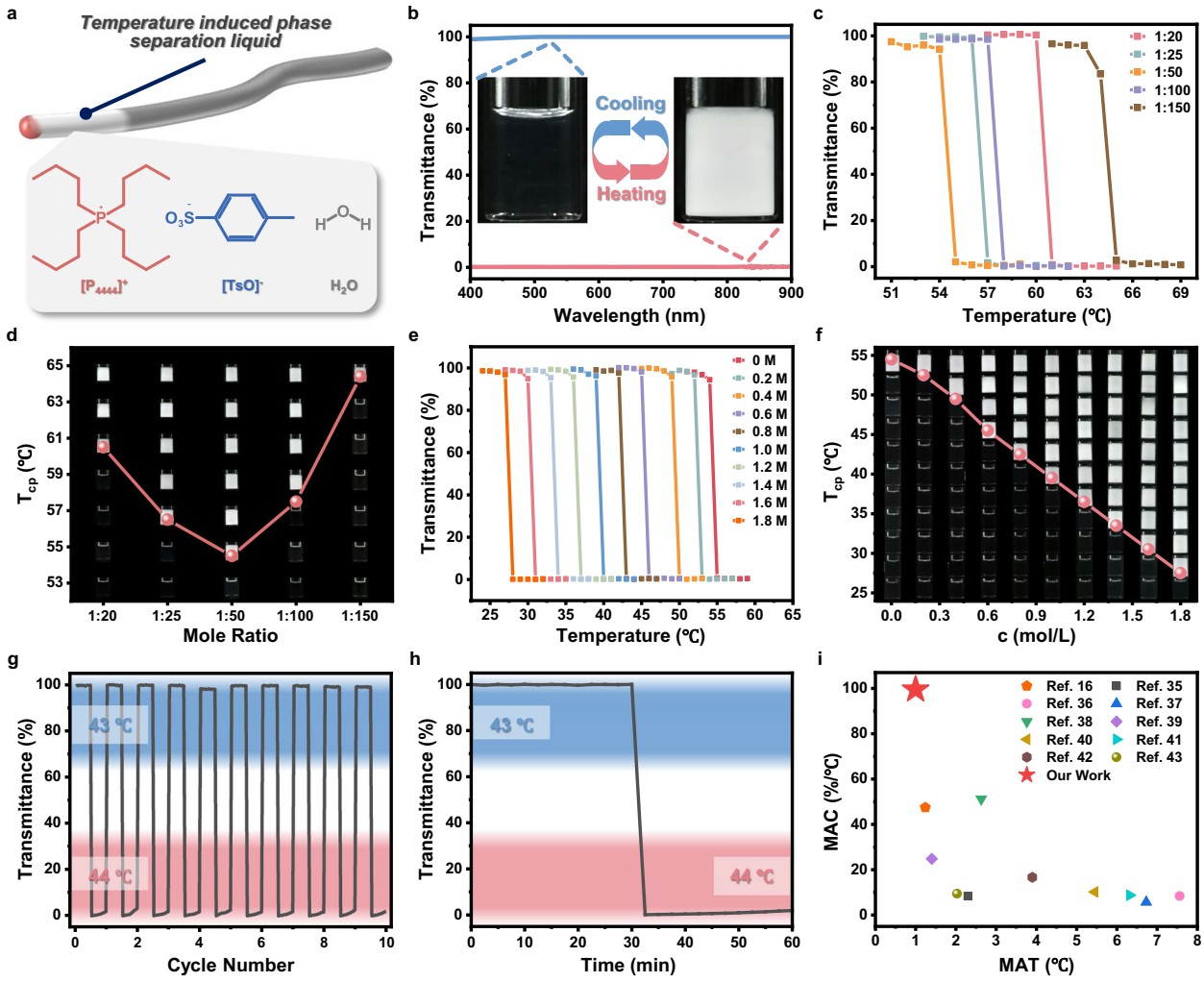

**Fig. 2 | Precisely adjusting the critical temperature of [P$_{4444}$][TsO] in H$_2$O for the preparation of thermal-responsive optical fiber. a** Schematic diagram of the optical fiber consisting of [P$_{4444}$][TsO] aqueous solution. **b** Light absorption spectra and optical photographs of the ionic liquid solution with [P$_{4444}$][TsO]:H$_2$O molar ratio of 1:20 before and after the macroscopic phase separation. **c** The transmittance at 600 nm as a function of the temperature for the aqueous solutions with different ionic liquid concentrations. **d** Cloud point temperatures and photographs of [P$_{4444}$][TsO] aqueous solutions with different [P$_{4444}$][TsO]:H$_2$O molar ratios. **e** The transmittance at 600 nm as a function of temperature for the ionic liquid solution with the [P$_{4444}$][TsO]:H$_2$O molar ratio of 1:50, upon stepwise addition of NaCl. **f** Cloud point temperature and photographs of the ionic liquid solutions with the [P$_{4444}$][TsO]:H$_2$O molar ratio of 1:50 by adding different concentrations of NaCl. **g** Demonstration of the reversible phase separation of the ionic liquid solution with the [P$_{4444}$][TsO]:H$_2$O molar ratio of 1:50 and the addition of 0.73 M NaCl. **h** Phase separation rate of an aqueous solution of ionic liquid with a molar ratio of 1:50 with the addition of 0.73 M NaCl. **i** An Ashby plot of comparing the thermal responsivity of [P$_{4444}$][TsO] with other materials as recorded by their MAC and MAT[16,35–43].

concentration or adding sodium chloride. Such an ionic liquid was transparent to the incident light but became opaque at temperatures above its LCST. All the issues of common concerns in the optical fiber systems are well solved, as illustrated in Fig. 1e: (I) Phototherapy can be actualized in deeper tissue sites; (II) Thermal protection is provided almost like an "on-off" switch based on a negative feedback modulation mechanism, without relying on irradiation power[33]; (III) Larger illumination area is promised with a short distance between the fiber outlet and the lesion.

## Results

### Thermo-responsive ionic liquid with adjustable phase-separation temperature

As shown in Fig. 2a, the key material for establishing AOWS is a thermal-responsive ionic liquid ([P$_{4444}$][TsO]), which was prepared via an ion exchange reaction between tetra-n-butylphosphonium bromide

([P$_{4444}$]Br) and sodium p-toluenesulfonate (Na[TsO]). Typically, two ionic compounds dissolved in the aqueous medium were repeatedly extracted by dichloromethane, driving the formation of the target ionic liquid of [P$_{4444}$][TsO] and transferring them into the organic phase. After the removal of dichloromethane, pure [P$_{4444}$][TsO] was collected and verified by $^1$H-NMR spectroscopy and mass spectrometry (Supplementary Figs. S4–S8). In particular, the [P$_{4444}$][TsO] aqueous solution exhibited excellent thermal sensitivity, from optically transparent to opaque as a result of the phase separation between [P$_{4444}$][TsO] and water at the temperature above LCST. At the [P$_{4444}$][TsO]:H$_2$O molar ratio of 1:20, the solvated ionic liquid was separated out of the solution and aggregated into microscopic droplets when the temperature rises above 61 °C (Supplementary Fig. S9). Accordingly, the solution transmittance decreased from 99.18% to 0.25% (Fig. 2b). Noting that the phase separation process was reversible, the solution was fully recovered to its transparent feature once the temperature fell

below 61 °C. As the control, neither of the two raw materials of $[P_{4444}]$Br and Na[TsO] exhibited thermal responsivity alike $[P_{4444}][TsO]$ (Supplementary Fig. S10).

To specify the factors affecting the phase separation, we defined a cloud point temperature ($T_{cp}$) as the temperature at which the transmittance of the $[P_{4444}][TsO]$-$H_2O$ binary system drops to 50% at 600 nm[34]. $T_{cp}$ is closely relevant to the $[P_{4444}][TsO]$ concentration as well as the addition of sodium chloride (NaCl). At the molar ratio of $[P_{4444}][TsO]$:$H_2O$ ranging from 1:20 to 1:150, $T_{cp}$ firstly decreased from 60.5 °C to 54.5 °C and then increased to 64.4 °C (Fig. 2c, d, Supplementary Fig. S11). At a given $[P_{4444}][TsO]$ concentration, $T_{cp}$ could also be adjusted by adding NaCl into the solution (Fig. 2e, f, Supplementary Fig. S12). For example, at the $[P_{4444}][TsO]$:water molar ratio of 1:50, $T_{cp}$ was decreased consistently from 54.5 °C to 27.5 °C upon stepwise addition of NaCl. It is almost linearly correlated with the NaCl concentration, particularly in the 0.6 M-1.8 M region. Following the way of changing $[P_{4444}][TsO]$:water ratio or adding NaCl, $T_{cp}$ could be precisely adjusted with high accuracy up to 1 °C (Supplementary Fig. S13).

Besides the high sensitivity in response to temperature change, the phase separation of $[P_{4444}][TsO]$ solutions was reversible against the cyclic heating and cooling processes (Fig. 2g, h). Two parameters to evaluate the phase separation materials in response to temperature change, including the maximum adjustable capacity (MAC) and the minimum adjustment temperature (MAT) (see more details in the Section "Method"), were also recorded for the $[P_{4444}][TsO]$ aqueous system, in contrast to some representative examples. Notably, the $[P_{4444}][TsO]$-based system possesses the largest MAC and the lowest MAT relative to other published formulas. Such an excellent thermal-responsive capability lays a solid foundation for imparting the temperature-gating function in the following preparation of AOWS[16,35–43] (Fig. 2i).

## Mechanism of the thermal-induced phase separation

$[P_{4444}][TsO]$ self-assembles in a colloid-like form in water according to the dynamic light scattering (DLS) analysis, which is consistent with the Tyndall phenomenon of the $[P_{4444}][TsO]$ aqueous solutions (Supplementary Fig. S14). The average hydration kinetic radius of the $[P_{4444}][TsO]$ aggregation is progressively increased while raising the temperature (Fig. 3a, Supplementary Fig. S15). This trend suggests the weakened association between ions and water molecules, along with the strengthened self-segregation at higher temperatures. As investigated by $^1H$ NMR spectroscopy (Fig. 3b), the proton signals ascribed to both the cation and the anion gradually shift to the low field with an interval of 0.062 ppm per 5 °C as the temperature rises from 20 °C to 60 °C, which should be a result of the gradual disassociation between $[P_{4444}][TsO]$ and water. In theory, a hydrogen bonding interaction is expected between the O atoms of $[TsO]^-$ and the H atoms of $H_2O$. It would be weakened at higher temperatures, corresponding to the enhanced shielding effect on the H atoms of the water molecule and an upfield shift of the water proton signal. In turn, the proton signal of $[P_{4444}][TsO]$ shows a downfield shift relative to the $D_2O$ signal at elevated temperatures. As control experiments, temperature-dependent $^1H$ NMR studies were performed using two non-protonic solvents, including chloroform-d and dimethylsulfoxide-d6 instead of $D_2O$ (Supplementary Figs. S16, S17). These two solvents cannot form a hydrogen bonding association with $[P_{4444}][TsO]$, so chemical shifts almost kept unchanged upon rising temperature in both cases. When $D_2O$ was replaced by $H_2O$, a lower $T_{cp}$ of the $[P_{4444}][TsO]$ solution was obtained since the hydrogen bonding interaction of $[TsO]^-$ with $D_2O$ is weaker than that with $H_2O$. In addition, a higher $T_{cp}$ of the $[P_{4444}][TsO]$ solution was obtained when urea was added as a hydrogen bonding competitor (Fig. 3c).

In another aspect, the addition of NaCl to the $[P_{4444}][TsO]$ aqueous solutions also leads to a downfield shift of the proton signals, suggesting that the hydrogen-bonding interactions between $[TsO]^-$

and water are perturbed by NaCl (Fig. 3d). According to the variable-temperature DLS analysis, increasing the concentration of NaCl and elevating the temperature have equivalent effects on the average hydration kinetic radius of the ionic liquids (Fig. 3e, f, Supplementary Fig. S18). Following these clues, a concept of "salt-temperature" equivalence was proposed to explain the role of temperature and salt in the regulation of $T_{cp}$ in $[P_{4444}][TsO]$ aqueous solutions. Typically, the sulfonic acid group in $[TsO]^-$ forms hydrogen bonds with water molecules, as presented by the hydrophilic feature, accounting for the transparent solution without the obvious phase separation. At higher temperatures, the hydrophilic effect is comprised as a result of the weakened hydrogen bonding, and accordingly, $[P_{4444}][TsO]$ prefers mutual aggregation to bonding water, ultimately leading to the increased degree of phase separation. Similarly, the addition of NaCl can attract those water molecules interacting with the ionic liquids due to their stronger hygroscopicity of smaller ions. Due to the salinization and electrostatic shielding effect upon NaCl addition, a higher degree of self-aggregation is predicted among the ionic liquids, similar to the result induced by elevated temperature (Fig. 3g).

## Design and optic assessment of AOWS

As profiled in Fig. 4a, AOWS for delivering light into deep tissues while having the ability of preventing overheating consists of an optical waveguide system (optical fiber), a light modulation system (a quartz capillary, a Luer connector, and a syringe), and a multifunctional fiber outlet (an elastic cap and silicone tube). The thermal-responsive ionic liquid aqueous solution, as stated above, was injected into the multifunctional capsule at the fiber tip through the light modulation system. Specifically, we chose an ionic liquid aqueous solution with the $[P_{4444}]$[TsO]:$H_2O$ ratio of 1:50 and the addition of 0.73 M NaCl as the waveguide gating material in the following investigations, unless otherwise notified, which has a $T_{cp}$ of 43.5 °C, close to the maximum safe temperature for biological tissues. A sudden decrease of light transmittance happens when the solution temperature rises above 43.5 °C owing to the separation of $[P_{4444}][TsO]$ from the water phase. Based on this performance, light propagation with high transmittance was allowed through the optical fiber when AOWS received a temperature below its $T_{cp}$, while it was impeded at temperatures above $T_{cp}$.

The optical fiber outlet at the end of AOWS was designed to have light-gating ability and adjustable divergence angles. It was composed of a rubber tube filled with an ionic liquid solution, which was sealed by an elastomer cap (Supplementary Figs. S19, S20). The longer tube length means the greater light scattering by the phase-separated ionic liquid solution at a temperature above $T_{cp}$ (Fig. 4b). Setting $P_0$ and $P_1$ as the optical power of the light passing through the ionic liquid solution without or with phase separation, over 99.8% ($P_1/P_0 = 0.1657\%$) of the incident light was notably scattered by the phase-separated liquid with a length of 1.0 cm (Fig. 4c). Similar to the above investigations, the ionic liquid solution sealed in the AOWS retains its reversible phase separation performance under alternate heating and cooling. In this regard, the AOWS can not only deliver interventional phototherapy to the deep tissue of interest, but also protect the tissue in the pathway of light propagation from overheating. Rsoft optical simulations further confirm the excellent optical transmission performance of AOWS in the absence of phase separation, regardless of the fiber outlet length (Supplementary Figs. S21, S22). The cap sealing the fiber outlet can be deformed depending on the amount of ionic liquid solution injected into the AOWS. It can be convex, flat, or concave with precise control over curvatures to match different divergence angles. As illustrated in Fig. 4d, a larger light-shedding area is obtained in the case of a concave cap relative to flat and convex shapes. It is predicted that AOWS can be adjusted with a desired light-shedding area while keeping it immobilized in the process of interventional phototherapy. These experimental observations are also theoretically supported by Rsoft's optical simulations (Fig. 4e, Supplementary Figs. S23, S24).

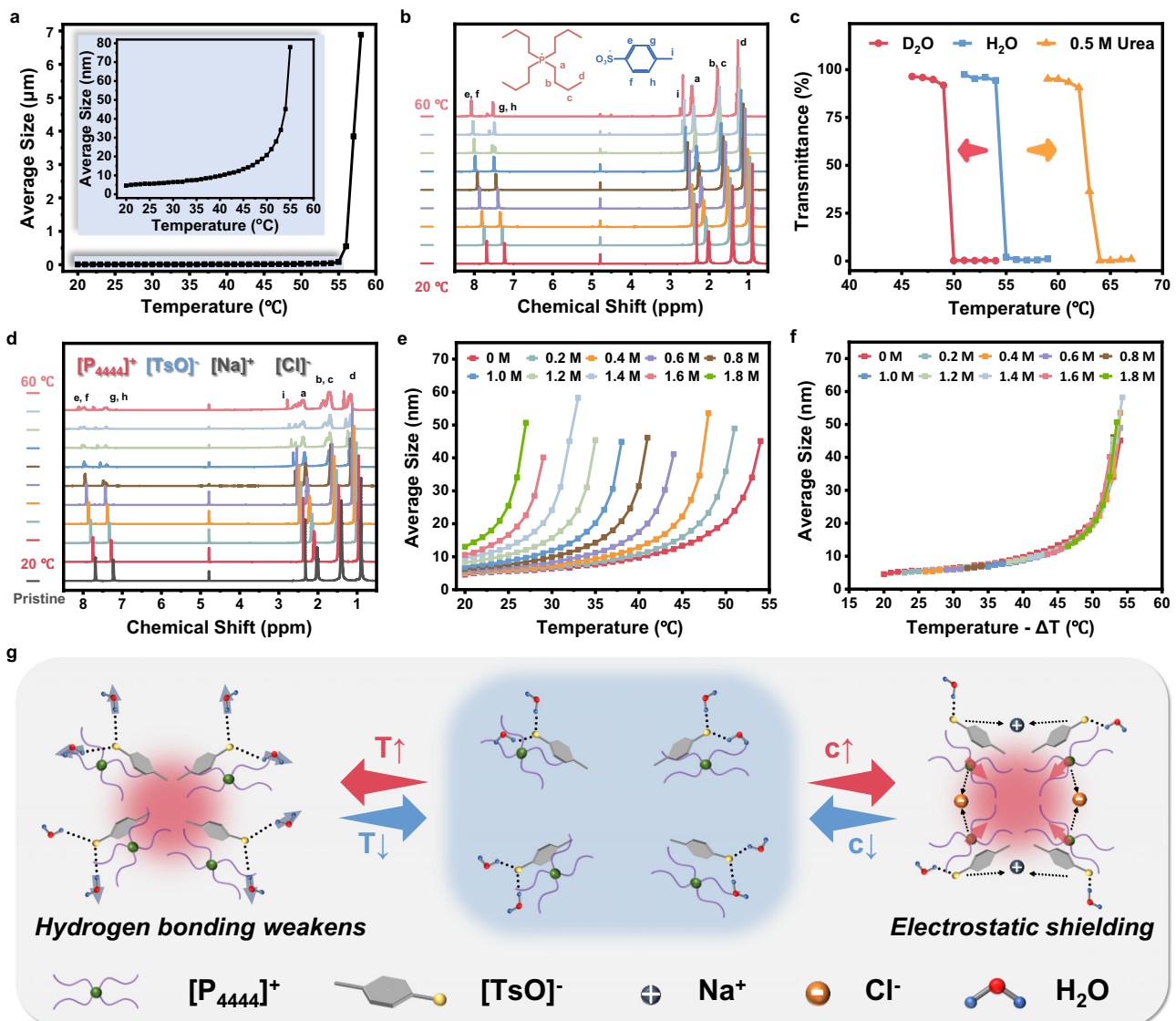

**Fig. 3 | Mechanisms of the phase separation upon increasing temperature or adding NaCl. a** Average size of ionic liquids with a molar ratio of 1:50 at different temperatures. The inset shows the detailed data in the 20–55 °C region. **b** Temperature-dependent $^1$H NMR spectra of [P$_{4444}$][TsO] in D$_2$O. **c** The transmittance at 600 nm as a function of temperature for the aqueous solution with the [P$_{4444}$][TsO]:H$_2$O molar ratio of 1:50, to which 0.5 M urea was added or not, and the deuterium aqueous solutions with the [P$_{4444}$][TsO]:D$_2$O molar ratio of 1:50.

**d** Temperature-dependent $^1$H NMR spectra of [P$_{4444}$][TsO] in D$_2$O with the addition of 0.8 M NaCl. **e** Average size of the aqueous solutions with the [P$_{4444}$][TsO]:H$_2$O molar ratio of 1:50 upon adding different concentrations of NaCl at different temperatures. **f** Average size of the aqueous solutions after "salt-temperature" equivalence. **g** Schematic illustration of the mechanism of the phase separation process in the circumstance of increasing temperature or adding NaCl.

## AOWS-tailored interventional phototherapy in simulated tissues

The controllable interventional phototherapy by AOWS was first verified in a simulated tissue made of cross-linked gelatin loaded with polypyrrole nanoparticles with photothermal conversion capabilities (Supplementary Fig. S25). As compared in Fig. 5a, the other two light-shedding models were supplied as the control groups, including light on skin (fiber cap outside the simulated tissue, 2.0 cm from the photothermal material) and the optical waveguide system (OWS) group (fiber cap inside the simulated tissue, 0.5 cm from the photothermal material). Based on the same apparatus, the OWS model differs from the AOWS model only in the use of water as a thermally insensitive waveguiding material in the optical fiber outlet. Then, the core temperature ($T_c$) of the hypothetical lesion tissue loading photothermal material and the optical fiber tip temperature ($T_a$) were recorded under three different light irradiation models (808 nm laser). Relative to the other two models, the AOWS exhibits excellent thermal protection performance with the premise of photothermal effect in the lesion tissue (Fig. 5b, Supplementary Figs. S26, S27). It shows a tendency towards negative feedback, indicating a self-regulating process in which both $T_a$ and $T_c$ are confined within a temperature range under continuous light input, unlike the continuous temperature rise in the OWS model. In the AOWS, when $T_a$ exceeds $T_{cp}$, the phase separation occurs in the optical fiber outlet, causing the incident light to be significantly scattered and unable to propagate further. Once $T_a$ is cooled below $T_{cp}$, the phase separation in the optical fiber outlet disappears, and light propagation is restored, followed by the next heating-cooling cycle. $T_c$ keeps pace with but is higher than $T_a$ owing to the photothermal effect of polypyrrole nanoparticles, ensuring a greater therapeutic effect on the lesion tissue but safer in the periphery normal tissue.

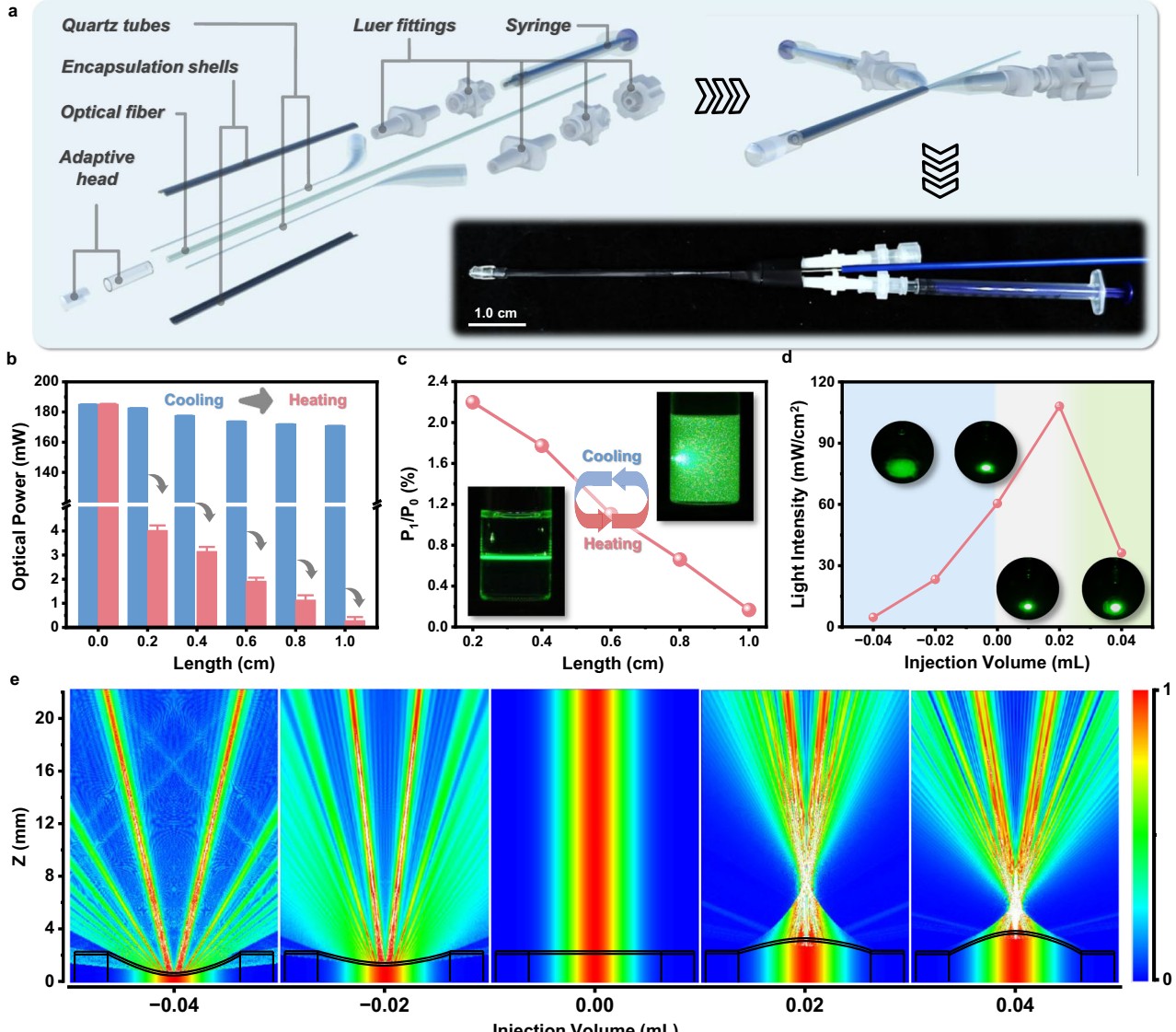

**Fig. 4 | Preparation and divergence angle regulation of AOWS. a** The components and an entity (insert photograph) of AOWS. **b, c** Light power decrement of a parallel incident light exiting AOWS using the fiber outlet with different light gating lengths before and after phase separation (*n* = 6, mean ± SD). **d** Light shedding area and power density of a parallel incident light exiting AOWS using optical fiber outlet with different features, either convex, flat, or concave with varying curvatures (*n* = 5, mean ± SD). **e** Simulation of light exiting through five fiber outlets with different curvatures, according to the experimental observations in (**d**).

The negative feedback capability of AOWS can prevent the overheating over a wide range of input light levels. As summarized in Fig. 5c, when $T_a$ is lower than $T_{cp}$ at lower incident light power, light propagation is always active in the absence of phase separation. Both $T_a$ and $T_c$ are far from causing a risk of overheating, although there is no negative feedback in AOWS. When the light power reaches 500 mW, the negative feedback control of temperature is triggered, allowing $T_a$ to oscillate within a small range due to the high thermal sensitivity of AOWS (Fig. 5c, Supplementary Fig. S28). Meanwhile, $T_c$ is maintained at no more than 55 °C, regardless of the input light power, which holds great promise for precise phototherapy without overheating diffusing into normal tissue. Adjusting the divergence angle by changing the convexity and concavity of the fiber cap provides us with a strategy for compromising the light power density and light irradiation area (Fig. 5d, e). Interventional phototherapy with sufficient irradiation area is assured even when the AOWS is close to the lesion tissue, which is impossible with traditional photytherapy strategies. Moreover, to strengthen the validity of our in vivo results, we performed ex vivo experiments on porcine muscle to further underscore the superiority of AOWS in achieving controlled photothermal therapy of subcutaneous tissues while preventing collateral overheating (Supplementary Fig. 29).

## Controllable photothermal performance of AOWS in vivo

Interventional phototherapy of the AOWS in deep tissue with thermal protection was further evaluated under in vivo conditions. As illustrated in Fig. 6a, AOWS was implanted into the hind leg muscle of a live pig as the experimental group, and the pig implanted with OWS served as the control group. Both AOWS and OWS were applied with an input light of a NIR laser with the wavelength of 808 nm at 900 mW, and their optical fiber tips were set at 0.4 cm from the photothermal material. As recorded by the thermocouple, the temperature at the fiber tip ($T_a$) and photothermal position ($T_c$) in the OWS group is distinctly higher than that in the AOWS group (Fig. 6b, Supplementary Fig. S30). The peripheral tissue around the photothermal core was further confirmed by the H&E staining to identify the phototherapy

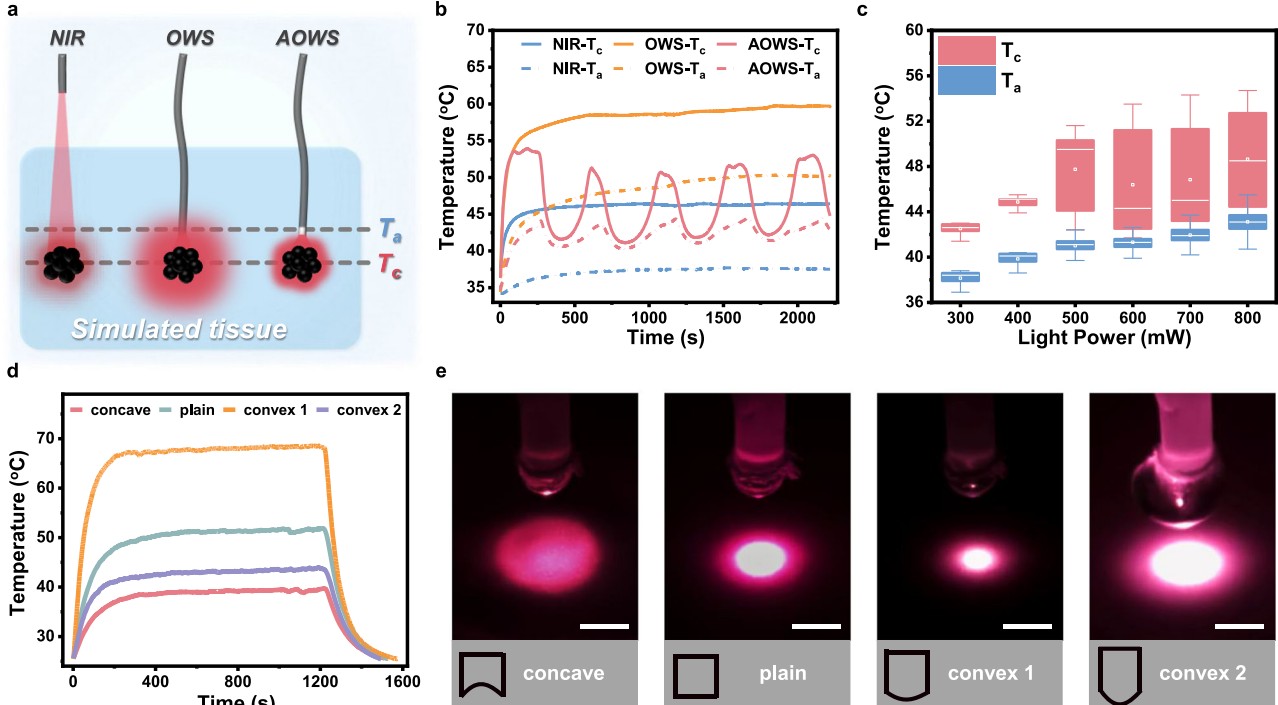

**Fig. 5 | Controlled photothermal performance of AOWS in simulated tissues.**
**a** Schematic illustration of three different photothermal models, including light on skin (NIR), optical waveguide system (OWS), and adaptable optical waveguide system (AOWS). $T_c$ and $T_a$ refer to temperatures at the core of the lesion tissue and the optical fiber tip. **b** Temperature rise of the photothermal material loaded into the lesion tissue ($T_c$) and the optical fiber tip ($T_a$) in three different photothermal models. **c** Temperature rise of the photothermal material loaded into the lesion tissue ($T_c$) and optical fiber tip ($T_a$) in the AOWS model using infrared light (808 nm) with different light powers. The horizontal line within the box represents the data value located in the middle sequence number, the upward bar represents the maximum data value, the downward bar represents the minimum data value, and the hollow dot within the box represents the average value. **d** Temperature of the photothermal material irradiated by the AOWS with the optical fiber caps having different convexity and concavity at a given input light power. **e** Photographs of the light shedding area by the AOWS with the optical fiber caps having different degrees of convexity and concavity. Scale bars: 2.0 mm.

effect and overheating extent of AOWS and OWS (Fig. 6c). In comparison, no thermal damage was observed in the tissue beyond 0.4 cm from the photothermal position for the AOWS group, whereas thermally induced coagulative necrosis was clearly observed at a distance of 0.8 cm from the photothermal material for the OWS group. These in vivo investigations are consistent with the benefit of AOWS for limiting the overheating effect within the lesion tissue. As explained by the simulation of thermal diffusion throughout the tissue, the higher temperature in the OWS group accounts for the overflow of hyperthermia into a larger spatial area (Fig. 6d, Supplementary Figs. S31, S32).

## Discussion
In summary, we established an optical waveguide system that could sense and respond to a specific temperature change at its optical outlet. This system included an ionic liquid whose aqueous solution could be adjusted from transparent to opaque at the desired temperature. Light propagation guided through the optical fiber was switched by the reversible phase separation of the ionic liquid solution, allowing control of the highest temperature in the light-shedding area. Another merit of the waveguide system is its adjustable divergence angle of the transmitted light, promising an adequate shedding area at a short distance from the optical fiber. This adaptive optical waveguide system was extended to the application of interventional phototherapy, which has demonstrated the benefits of negative feedback capability and sufficient irradiation area in the lesioned tissue. It is believed to be a powerful adjunct or alternative to the clinically applied electrothermal therapy technology of radiofrequency ablation, in

order to minimize the risk of overheating and undesirable damage to normal tissue.

## Methods
### Materials
Sodium chloride, gelatine (~250 g bloom) and dimethyl sulfoxide-d6 were purchased from Meryer Chemical Technology Co., Ltd. (Shanghai, China). Tetrabutylphosphonium bromide ([$P_{4444}$]Br) was obtained from Shanghai Macklin Biochemical Technology Co., Ltd. (Shanghai, China). Polyvinyl alcohol (PVA 1788) and potassium persulfate (KPS) were supplied by Shanghai Aladdin Biochemical Technology Co., Ltd. (Shanghai, China). Sodium p-toluenesulfonate (Na[TsO]) was obtained from Tianjin Hiens Biochemical Technology Co., Ltd. (Tianjin, China). Pyrrole was purchased from Beijing InnoChem Science & Technology Co., Ltd. (Beijing, China). Glutaraldehyde was supplied by Shanghai Energy Chemicals Co., Ltd. (Shanghai, China). Dichloromethane was purchased from Beijing Tong Guang Fine Chemicals Company (Beijing, China). Deionized water was obtained by a Milli-Q water purification system. SMA905 commercial optical fiber was purchased from Shenzhen Xinrui Photonics Technology Co., Ltd. (Shenzhen, China). Customized quartz capillary tubes were provided by Lianyungang Gongxing Quartz Products Co., Ltd. (Jiangsu, China). Luer fittings were supplied by Shenzhen Jiawei Medical Technology Co., Ltd. (Shenzhen, China). GCVW-NIR3 Infrared Detection Card was purchased from Daheng New Era Technology Co., Ltd. (Beijing, China). The silicone rubber tubing was obtained from Taobao.com. The transparent elastomer was a nano double-sided adhesive provided by M&G Chenguang Stationery Co., Ltd. (Shanghai, China).

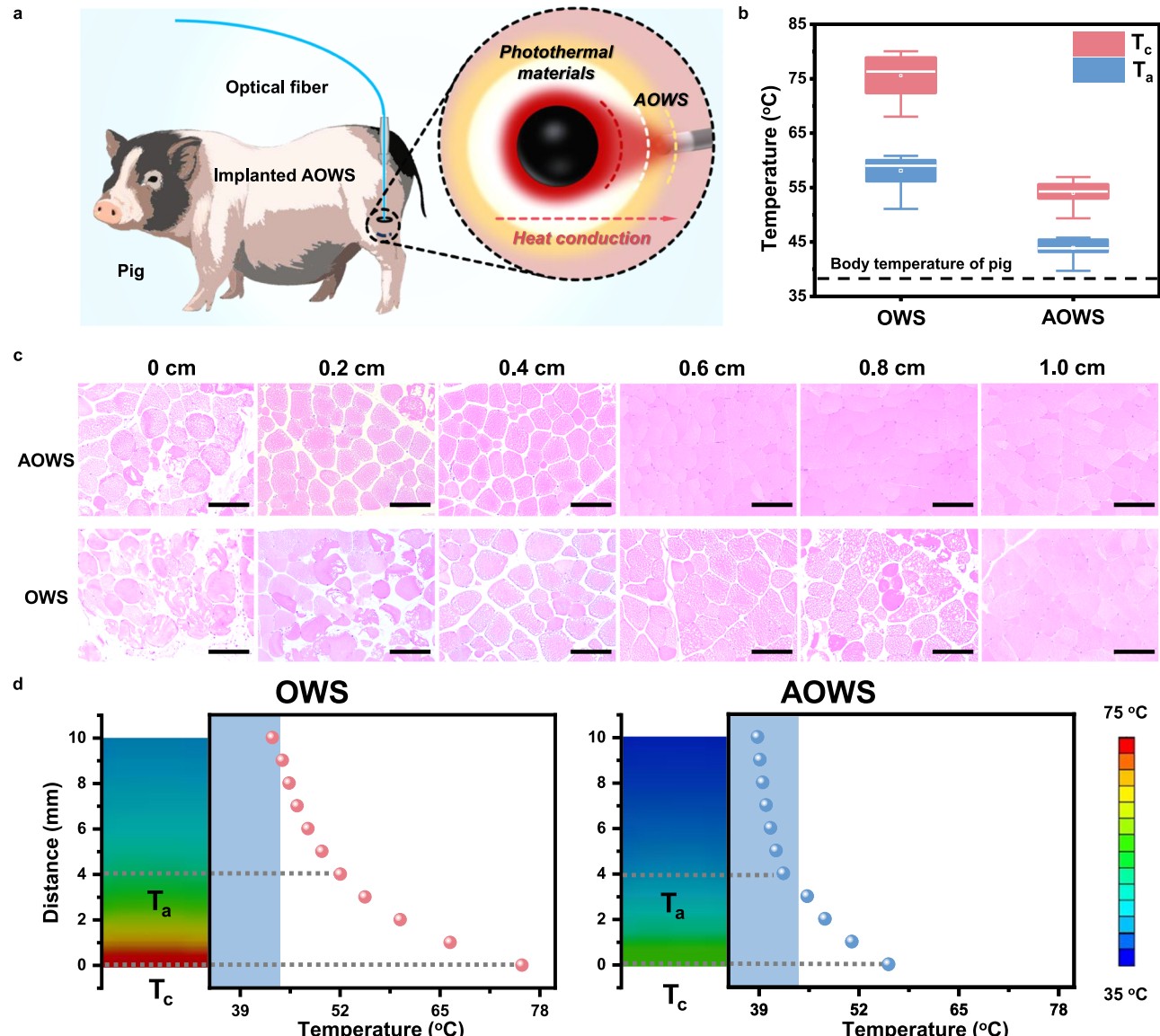

**Fig. 6 | Evaluation of photothermal performance of AOWS under in vivo conditions. a** Schematic illustration of the interventional phototherapy on a live pig, either in AOWS or OWS models. **b** Temperature at the photothermal material ($T_c$) and the optical fiber tip ($T_a$) in AOWS and OWS experimental groups. The horizontal line within the box represents the data value located in the middle sequence number, the upward bar represents the maximum data value, the downward bar represents the minimum data value, and the hollow dot within the box represents the average value. **c** Representative H&E staining results of the tissue at difference distance from the photothermal core ($n = 3$). Scale bars: 10.0 μm. **d** The simulation of thermal diffusion throughout the tissue in AOWS and OWS experimental groups. The blue-shaded area is the safe temperature range.

## Characterization

The thermostatic magnetic stirrer (LICHEN, 101S) was used to heat and mix the solutions. The average hydrodynamic diameter of [$P_{4444}$][TsO] aqueous solution and polypyrrole nanoparticles (PPy NPs) was measured by the Zetasizer Nano ZS90 (Malvern Panalytical, Malvern, United Kingdom). All UV-Vis-NIR spectra of solutions were obtained on a Shimadzu UV-3600 spectrometer with a temperature control attachment (Shimadzu Corp., Kyoto, Japan). All the refractive indices of liquids were measured by a WAY Abbe refractometer (2WAJ, Shanghai Precision Scientific Instrument Co., Ltd.). $^1$H NMR and $^{13}$C NMR spectra were measured by a Bruker 400 MHz NMR spectrometer, and temperature-dependent $^1$H NMR spectra were measured by a Bruker 600 MHz NMR spectrometer (Bruker Corporation, USA). High-resolution electrospray ionization mass spectra were measured by a Shimadzu LCMS-IT/TOF mass spectrometer (Shimadzu Corp., Kyoto, Japan). The power density of the laser was detected by a digital handheld optical power and energy meter (Thorlabs, PM100D; Thorlabs Inc., Newtown, NJ, United States). The 532 nm and 808 nm laser source was supplied by Hi-Tech Optoelectronics Co., Ltd. The thermocouple (UNI-T, UT325F) with an ultra-fine probe was utilized to record the photothermal temperature of the photothermal conversion agents. An indium-tin-oxide glass-based temperature control platform was used to regulate the temperature in taking microscopic photographs of the solutions. Color images of tissue section staining were captured by the DMi8 inverted fluorescence microscope (Leica, Wetzlar, Germany). Temperature recorded by infrared camera (Tix640, Fluke) during photothermal efficiency test. The fiber tip modulus was characterized using a dynamic mechanical analyzer (DMA 850, TA Instruments). All optical images of noted experiments in the work were captured by a Canon EOS 70D digital camera.

## Preparation of [P$_{4444}$][TsO]

[P$_{4444}$][TsO] was synthesized through an ionic exchange reaction. Typically, 20.00 g of [P$_{4444}$]Br was dissolved in 100.0 mL of deionized water, followed by the addition of 11.44 g of Na[TsO]. Na[TsO] was slightly excessive relative to [P$_{4444}$]Br to ensure the complete consumption of [P$_{4444}$]Br. The mixture was stirred overnight and ended up with a pale-yellow solution. Subsequently, dichloromethane (CH$_2$Cl$_2$) was used to extract [P$_{4444}$][TsO] from the aqueous phase and washed with deionized water to completely remove the impurities. Pure [P$_{4444}$][TsO] was obtained after removing the CH$_2$Cl$_2$ through rotary evaporation. $^1$H NMR and $^{13}$C NMR spectra are as follows:

$^1$H NMR (400 MHz, D$_2$O) δ 7.72–7.66 (s, 2H), 7.39–7.35 (s, 2H), 2.40 (s, 3H), 2.20–2.05 (m, 8H), 1.59–1.38 (m, 16H), 0.92 (t, 12H).

$^{13}$C NMR (100 MHz, D$_2$O) δ 142.36, 139.56, 129.41, 125.36, 23.34, 23.19, 20.48, 17.84, 17.36, 12.53.

## The transmittance measurement of [P$_{4444}$][TsO] aqueous solutions

The transmittance of [P$_{4444}$][TsO] aqueous solutions was recorded on a UV-Vis-NIR spectrophotometer with a temperature control attachment. Before each measurement, the [P$_{4444}$][TsO] aqueous solutions were stabilized for 15 min to allow for complete temperature equilibration. T$_{cp}$ was identified as the temperature at which the transmittance at 600 nm is 50%.

## Calculation of the divergence angle of commercial optical fibers

The commercial fiber is a multi-mode energy transmission fiber that consists of a core with a higher refractive index (n$_1$) and a cladding with a lower refractive index (n$_2$). Due to the relatively small difference between n$_1$ and n$_2$, the International Telegraph and Telephone Consultative Committee (CCITT) recommends in ITU-T G.651 that the numerical aperture of multimode fibers be taken in the range of 0.18 to 0.23 (Eq. (1)). According to the definition of NA (Eq. (2)), the divergence angle of commercial optical fibers in water (n = 1.33) is ±7.74° - ±9.96°.

$$NA = \sqrt{n_1^2 - n_2^2} \tag{1}$$

$$NA = n \sin \alpha \tag{2}$$

## Definition of MAC and MAT

MAC reflects the ability to modulate light before and after the phase separation. Therefore, MAC is defined as,

$$MAC = \frac{T_{max}\% - T_{min}\%}{|T_{max} - T_{min}|} \tag{3}$$

where T$_{max}$% and T$_{max}$ represent the maximum value of light transmission and its corresponding temperature, respectively; T$_{min}$% and T$_{min}$ represent the minimum value of light transmission and its corresponding temperature, respectively. MAT stands for the minimum adjustable accuracy of T$_{cp}$. Therefore, MAT is defined as the minimum temperature gradient over which the T$_{cp}$ can be adjusted. Due to the large variation in adjustable temperature gradients in some of the references, the average temperature gradient was used as their MAT.

## Device simulation

The beam propagation method (BPM) in RSoft, a commercial photonic simulation software, was used to simulate the light propagation of the AOWS under different states of convexity and concavity.

## Synthesis of PPy NPs

PPy NPs were synthesized by the emulsion method. 10.0 mL of water containing 0.5 wt.% polyvinyl alcohol (PVA 1788) was used as the aqueous phase, and the oil phase was 2.0 mL of methylene chloride (DCM) dissolving 100.0 μL of pyrrole. Oil-in-water emulsions were produced by a high-speed disperser, followed by adding 0.40 mL of KPS aqueous solution (2.0 M) to initiate the polymerization of pyrrole in the oil phase. Then the DCM was removed by volatilization. Finally, the polypyrrole nanoparticles (PPy NPs) dispersion was obtained after dialysis purification.

## Preparation of simulated tissues

To fabricate the simulated tissue model, 5.0 mL of a 10 wt.% gelatin solution was dispensed into a Petri dish (3.50 cm diameter) and crosslinked by the addition of 100 μL of 10 wt.% glutaraldehyde. Crosslinking was carried out at 4 °C for 24 h to ensure uniform network formation. Thereafter, a central cavity (Ø0.80 cm) was created using a biopsy punch, and 0.40 mL of 10 wt.% gelatin solution containing 1.0 mg·mL$^{-1}$ polypyrrole was introduced into the cavity. The composite construct was then subjected to the same 4 °C, 24 h glutaraldehyde crosslinking protocol to yield the final simulated tissue.

## Preparation of AOWS

Luer fittings were separately inserted into two custom quartz tubes of similar length. And then 502 glue was applied to reinforce the joints. After that, these two tubes were glued in parallel on both sides of the SMA905 commercial optical fiber. Finally, the AOWS was prepared after attaching a 1.0 cm silicone rubber tube to the head of the fiber and encapsulating it with a transparent elastomer.

## Controlled photothermal performance of AOWS on living pigs

The 10-11-month-old Bama miniature pigs (females) were provided by Beijing Sitronix Miniature Pig Breeding Base and kept in the animal laboratory of Beijing Institute of Traumatology and Orthopedics. All animal experiments performed in this study were in accordance with the protocol approved by the Experimental Animal Ethics Committee of the Beijing Institute of Traumatology and Orthopedics and were conducted in accordance with the Regulations for the Administration of Affairs Concerning Experimental Animals approved by the State Council of the People's Republic of China. The license number of the facility for the use of laboratory animals is SYXK (Beijing) 2023–2021. Firstly, the hair of anesthetized experimental pigs was shaved. Secondly, polypyrrole-dried gelatin sheets attached with ultrafine thermocouples and AOWS were implanted subcutaneously in the muscle of the hind leg. Finally, a laser with a wavelength of 808 nm was introduced into AOWS while T$_a$ and T$_c$ were measured and recorded by UT325F.

## Statistics analysis

The data of interest was processed by Microsoft Office 2021 Excel. Temperature data was recorded by a thermocouple (UT325F). Data was made into graphs by OriginLab 2024 (student edition).

## Reporting summary

Further information on research design is available in the Nature Portfolio Reporting Summary linked to this article.

# Data availability

All data supporting the findings of this study are available within the article and its supplementary files. Any additional requests for information can be directed to and will be fulfilled by the corresponding author. Source data are provided with this paper.

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

## Acknowledgments

We thank Prof. Tairan Fu (Department of Energy and Power Engineering, Tsinghua University) for his support in the analysis of the finite element analysis. This work was financially supported by the National Natural Science Foundation of China (22335008). All animal experiments per-formed in this study were in accordance with the protocol approved by the Experimental Animal Ethics Committee of the Beijing Institute of Traumatology and Orthopedics and were conducted in accordance with the Regulations for the Administration of Affairs Concerning

Experimental Animals approved by the State Council of the People's Republic of China.

## Author contributions

Z. Wang, Z. Yang, Y. Ma and X. Hu performed the research under the supervision by Prof. Y. Wang. S. Lyu performed the analysis of the heat transfer model and FEMs. Z. Wang, Z. Yang, Y. Ma and Y. Wang wrote the manuscript.

## Competing interests

Y. Wang, Z. Wang, Z. Yang and Y. Ma are coinventors on a patent for large-area and overheating-preventing phototherapy in deep tissue. The remaining authors declare no competing interests.
