## [Transparent Peer Review file · Nature Communications]

Adaptive optical waveguide system for large-area and overheating-preventing phototherapy in deep tissue

Corresponding Author: Professor Yapei Wang

Version 1:

Reviewer comments:

Reviewer #1

(Remarks to the Author)

The author has reported a fiber optic device applicable for temperature control in deep-tissue photothermal cancer therapy, and detailed experimental verifications have also been conducted. Nevertheless, the novelty of this concept within the article is not clearly expounded. Specifically, there have been previous reports on deep photothermal cancer therapy with fiber optic intervention and its temperature control (e.g., Nature Communications 13 (1), 7789; Advanced Fiber Materials 5 (3), 968-978), and although the author has cited some of these articles, they are not appropriately placed for a direct description and are instead evasively skipped over. Before the article can be accepted, the author must clearly elaborate on the distinctions between the innovations presented in this paper and the concepts described in the existing several articles (ACS Materials Lett. 2024, 6, 4854–4864; National Science Review 8 (9), nwaa209; Chinese Chemical Letters 2025, doi.org/10.1016/j.ccl.2024.110440), rather than subjectively considering the concept proposed herein as novel. It is recommended that the author undertake a detailed review of the existing fiber optic-intervened photomedicine treatments and clearly analyze the innovation points of this article. This innovation point does not necessarily have to be the controllability of temperature in deep-tissue photothermal cancer therapy and thus requires further refinement. And the following issues are also provided for reference:

Question 1: Whether the real-time temperature variations at the tumor site can be presented in animal experiments could directly demonstrate the feasibility of temperature control by the optical fiber temperature control system in practical applications.

Question 2: The manuscript describes regulating the divergence angles by adjusting different types of optical fiber outlets, it is a very interesting point. However, it would be even better if there were specific data to support the optical transmission efficiency under various outlet configurations.

Question 3: The format of the references still needs to be standardized, such as the issue of capitalization in titles, etc.

Reviewer #2

(Remarks to the Author)

Reviewer #3

(Remarks to the Author)

Phototherapy, valued for its non-toxicity, selectivity, and minimal trauma, is predominantly applied to treat superficial diseases due to the limited penetration of light through tissues. While optical-fiber-assisted interventional phototherapy addresses this limitation, it lacks an immediate mechanism to mitigate overheating of surrounding healthy tissues. This manuscript introduces an adaptive optical waveguide system (AOWS) based on a thermoresponsive ionic liquid with a tunable lower critical solution temperature (LCST), which facilitates deep-tissue phototherapy while mitigating overheating of surrounding healthy tissues through a negative feedback mechanism. The system addresses critical challenges in conventional phototherapy, such as extended light paths and inadvertent thermal damage to non-target tissues, thereby offering a safer and more efficient solution for deep-tissue applications. The data and characterization presented robustly support the conceptual framework and system design of the authors. Given the novelty and comprehensive characterizations

of the proposed system, I recommend its publication in Nature Communications following minor revisions.

1. The authors assert that the fiber employed in the system protects biological tissues by thermally safeguarding their structure; however, biological tissues are also sensitive to external mechanical stresses, such as friction. It is unclear whether the fiber tip is rigid or flexible, and it would be valuable to include tests of the mechanical properties of the fiber tip, such as its modulus.
2. The paper describes the development of an adaptive optical waveguide system (AOWS) for deep phototherapy with tissue protection, leveraging a negative feedback modulation mechanism. While this concept is intriguing, the authors do not clearly define the mechanism or compare its advantages to conventional thermal protection strategies. A more detailed discussion of the concept, along with additional evidence supporting its benefits, would strengthen the manuscript and clarify this novel approach.
3. The "simulated tissue" used in the study (cross-linked gelatin with polypyrrole nanoparticles) lacks explicit synthesis parameters, such as the size distribution of the nanoparticles. Furthermore, the photothermal conversion efficiency of the photothermal material is a critical parameter that should be measured and reported, as it plays a vital role in evaluating the performance of the system in thermal protection for photothermal therapy.

Version 2:

Reviewer comments:

Reviewer #3

(Remarks to the Author)

The current version after revision could be accepted as is.

Reviewer #4

(Remarks to the Author)

The manuscript presents an adaptive optical waveguide system (AOWS) that employs an ionic-liquid aqueous solution with a tunable lower critical solution temperature (LCST). Above a preset temperature the solution undergoes a reversible, rapid phase transition to an opaque state, producing a temperature-triggered negative-feedback gating mechanism that scatters transmitted light. This mechanism mitigates overheating of surrounding healthy tissue while preserving effective photothermal treatment at the target site. Unlike conventional passive thermal management or externally driven modulation, the proposed strategy implements an autonomous, local safety control that is conceptually novel and practically deployable for photothermal therapy. Experimentally, the system achieves near 'on-off' thermal protection that is independent of irradiation power. Moreover, a tailored fiber-optic tip permits controlled adjustment of beam divergence, enabling large-area subcutaneous illumination with tunable treatment radii. In addition, comprehensive material characterization, quantifiable temperature-gating behavior, and the divergence-angle design of the manuscript are corroborated by thermal and optical simulations as well as ex-/in-vivo demonstrations, which together substantiate the mechanism and therapeutic efficacy. High-quality figures and a rigorous, quantitative presentation further strengthen the credibility of the proposed mechanism and the conclusions. Therefore, I strongly recommend its publication in Nature Communications.

Replies to Reviewers' comments and questions

Reviewer #1: The author has reported a fiber optic device applicable for temperature control in deep-tissue photothermal cancer therapy, and detailed experimental verifications have also been conducted. Nevertheless, the novelty of this concept within the article is not clearly expounded. Specifically, there have been previous reports on deep photothermal cancer therapy with fiber optic intervention and its temperature control (e.g., Nature Communications 13 (1), 7789; Advanced Fiber Materials 5 (3), 968-978), and although the author has cited some of these articles, they are not appropriately placed for a direct description and are instead evasively skipped over. Before the article can be accepted, the author must clearly elaborate on the distinctions between the innovations presented in this paper and the concepts described in the existing several articles (ACS Materials Lett. 2024, 6, 4854–4864; National Science Review 8 (9), nwaa209; Chinese Chemical Letters 2025, doi.org/10.1016/j.ccllet.2024.110440), rather than subjectively considering the concept proposed herein as novel. It is recommended that the author undertake a detailed review of the existing fiber optic-intervened photomedicine treatments and clearly analyze the innovation points of this article. This innovation point does not necessarily have to be the controllability of temperature in deep-tissue photothermal cancer therapy and thus requires further refinement. And the following issues are also provided for reference:

[Reply to Reviewer #1] Thank you very much for your time and efforts in reviewing this manuscript, your suggestions and comments give us inspirations to further improve the work. Our point-to-point responses to all your concerns are as below:

Comment: The novelty of this concept within the article is not clearly expounded. Specifically, there have been previous reports on deep photothermal cancer therapy with fiber optic intervention and its temperature control (e.g., Nature Communications 13 (1), 7789; Advanced Fiber Materials 5 (3), 968-978), and although the author has cited some of these articles, they are not appropriately placed for a direct description and are instead evasively skipped over. Before the article can be accepted, the author must clearly elaborate on the distinctions between the innovations presented in this paper and the concepts described in the existing several articles

(ACS Materials Lett. 2024, 6, 4854–4864; National Science Review 8 (9), nwa209; Chinese Chemical Letters 2025, doi.org/10.1016/j.ccllet.2024.110440), rather than subjectively considering the concept proposed herein as novel. It is recommended that the author undertake a detailed review of the existing fiber optic-intervened photomedicine treatments and clearly analyze the innovation points of this article. This innovation point does not necessarily have to be the controllability of temperature in deep-tissue photothermal cancer therapy and thus requires further refinement.

[Reply to Comment]: Thank you for comment on highlighting the need to more clearly define our feedback-modulation mechanism and to position it relative to existing thermal-protection strategies. We have supplemented the following discussion to clearly compare its advantages to conventional thermal protection strategies.

In recent years, photothermal therapy (PTT) has progressed toward precise temperature monitoring and autonomous thermal control. For accurate temperature monitoring, Wei et al. adjusted irradiation power density and exposure duration, employing repeated irradiation cycles to maintain therapeutic efficacy while avoiding thermal injury (*Adv. Healthcare Mater.* **2023**, 12, 2202360). Zheng et al. developed a photothermoelectric cobalt-infused chitosan nanocomposite hydrogel that exhibits a linear correlation between temperature shifts and resistance changes, enabling precise subcutaneous temperature measurements to prevent overheating (*Adv. Healthcare Mater.* **2024**, 13, 2401609). However, these approaches cannot respond instantaneously to thermal excursions and instead rely on manual adjustments of laser power based on real time temperature feedback.

In the area of autonomous temperature regulation, Zhu et al. proposed a photomedicine strategy using a temperature adaptive hydrogel optical fiber waveguide (*Nat. Commun.* **2022**, 13, 7789). This strategy harnesses the phase transition of a responsive material as an optical switch to limit the core temperature of the photothermal agent. Because the fiber tip and the photothermal medium are in intimate contact, thermal equilibrium is reached rapidly. In practice, however, variations in tissue thermal conductivity and laser power produce substantial differences in the resulting steady state thermal distribution and ablation volume. Ensuring that the treated region precisely matches the lesion geometry therefore requires extremely fine

tuning of both the phase transition temperature and the laser output. Without such precision, neighboring healthy tissues remain vulnerable to unintended thermal damage, a challenge that is difficult to surmount in clinical applications.

In our design, the photothermal material is separated by a small gap from the fiber tip, which is loaded with the phase-change material, rendering heat transfer the rate-limiting step. So, the negative feedback mechanism goes like this: when the tip temperature exceeds the phase-change threshold, scattering within the phase-change medium blocks the optical path, arresting further heating and safeguarding healthy tissue; conversely, if the tip temperature remains below this threshold, optical transmission continues unabated, allowing the lesion temperature to rise and induce thermal coagulative necrosis. Because this mechanism relies on a dynamically evolving thermal field rather than a fixed steady-state, its effectiveness is essentially independent of laser power. It requires only that the phase-change temperature be set between the tolerance limits of normal and target tissues and positioned at the edge of the intended ablation zone. By limiting the thermal kill region to these bounds, our approach greatly reduces the risk of unintended thermal injury to surrounding healthy tissue.

Overall, through its feedback-based regulation mechanism, AOWS can provide thermal protection that operates almost like an **"on-off" switch, without relying on irradiation power.**

Thank you again for your thoughtful suggestion. We have cited all the literatures you provided and carefully refined the introduction section of the main text. And all manuscript revisions marked with yellow background.

Q1: Whether the real-time temperature variations at the tumor site can be presented in animal experiments could directly demonstrate the feasibility of temperature control by the optical fiber temperature control system in practical applications.

[Reply to Q1]: Thank you for your insightful suggestion. Actually, the adaptive optical waveguide system (AOWS) was developed to enable large-area phototherapy of deep-seated pathological tissues (such as tumors and nodules) while preventing local overheating. Although our focus includes but is not limited to neoplastic lesions, prior studies have shown that normal cells exhibit greater thermal tolerance than tumor cells: irreversible damage to tumor cells

occurs above 42 °C, whereas normal cells incur irreversible injury only above 46 °C (*Front. Bioeng. Biotechnol.* **2024**, 12, 1432189.). Accordingly, in our animal experiments, we designated the core temperature of the photothermal material (T_c) as the lesion temperature and the temperature at the fiber tip (T_a) as the boundary temperature of adjacent healthy tissue, and we continuously recorded the real-time temperature variations at both locations. As illustrated in Figure R1, AOWS was implanted into the hind-leg muscle of live pigs (experimental group), while conventional optical waveguide system (OWS) served as controls. Both groups were irradiated with an 808 nm NIR laser at 900 mW, with fiber tips positioned 0.4 cm from the photothermal material. Thermocouple measurements revealed that T_a and T_c in the OWS group were significantly higher than in the AOWS group (Figure R1b, c). Hematoxylin and eosin staining of perilesional tissue (Fig. 6c in manuscript) confirmed that no thermal damage occurred beyond 0.4 cm from the photothermal core in the AOWS group, whereas coagulative necrosis extended to 0.8 cm in the OWS group. These in vivo results align with our thermal-diffusion simulations, which attribute the wider hyperthermia spread in the OWS group to uncontrolled heat overflow (Figure R1d).

Figure R1. (a) Photograph of the live pig photothermal experimental device. (b) Temperature at the photothermal material (T_c) and the optical fiber tip (T_a) in the experimental group

implanted with AOWS. (c) Temperature at the photothermal material (T_c) and the optical fiber tip (T_a) in the control group implanted with OWS. (d) The simulation of thermal diffusion throughout the tissue in AOWS and OWS experimental groups. The blue-shaded area is the safe temperature range.

Moreover, to strengthen the validity of our *in vivo* results, we performed *ex vivo* experiments on porcine muscle, monitoring real-time temperature dynamics (Figure R2). Under 808 nm, 800 mW irradiation, the AOWS group demonstrated robust negative-feedback regulation of both the fiber-tip temperature (T_a) and the core temperature (T_c). Importantly, T_c exceeded the threshold required for effective lesion cell ablation, while T_a remained within the thermal tolerance range of healthy tissue. By contrast, the OWS group exhibited markedly elevated T_c and T_a values, with T_a surpassing the tolerance limit of normal tissue and therefore posing a risk of inadvertent thermal injury. These *ex vivo* findings further underscore the superiority of AOWS in achieving controlled photothermal therapy of subcutaneous tissues while preventing collateral overheating.

Figure R2. (a) Diagram of the setup for the isolated pork experiment. (b) Temperature at the photothermal material (T_c) and the optical fiber tip (T_a) in AOWS and OWS experimental groups.

Q2: The manuscript describes regulating the divergence angles by adjusting different types of optical fiber outlets, it is a very interesting point. However, it would be even better if there were specific data to support the optical transmission efficiency under various outlet configurations.

[Reply to Q2]: Thank you for your professional and forward-thinking comments. Based on your suggestions we have supplemented the data on divergence angle and optical transmission efficiency (OTE) at 532 nm and 808 nm with experiments and optical simulations.

First, to ensure maximum accuracy in our optical simulations, we obtained a precise measurement of the refractive index of the aqueous ionic liquid at 532 nm. Because the Abbe refractometer only provides data at 589.3 nm, we instead employed the minimum deviation angle method (*Korean J. Opt. Photon.*, **2008**, 19, 182–186).

Figure R3. Schematic diagram of the minimum deviation angle method for measuring the refractive index of liquids.

In this approach, a hollow triangular prism (apex angle α) is filled with liquid and illuminated by a 532 nm laser along the DE face; the emergent beam exits the FG face. We define the total angular deviation δ between the incident ray DE and the emergent ray FG. From simple geometry:

$$\delta = \angle 1 + \angle 2 = (i_1 - i_2) + (i_1' - i_2') = (i_1 + i_1') - (i_2 + i_2') \quad (\text{R1})$$

$$i_2 + i_2' + (\pi - \alpha) = \pi \quad (\text{R2})$$

Equations R1 and R2 yield:

$$\alpha = i_2 + i_2' \quad (\text{R3})$$

$$\delta = i_1 + i_1' - \alpha \quad (\text{R4})$$

By Snell's law at faces AB and AC:

$$\sin i_1 = n \sin i_2 \quad (\text{R5})$$

$$\mathbf{sin}i_1' = n\mathbf{sin}i_2' \quad (\text{R6})$$

For a certain value of i_1 , δ has a minimum value of δ_{min} called the minimum angle of deflection, which can be used for the measurement of the refractive index of transparent materials. At the minimum-deviation condition, δ attains its minimum δ_{min} , when

$$\frac{d\delta}{di_1} = \mathbf{1} + \frac{di_1'}{di_1} = \mathbf{0} \quad (\text{R7})$$

$$\frac{di_1'}{di_1} = \mathbf{-1} \quad (\text{R8})$$

Differentiate Equation R3 with respect to i_2' gives:

$$\frac{di_2}{di_2'} = \mathbf{-1} \quad (\text{R9})$$

and differentiating Snell's law:

$$\mathbf{cos}i_1 di_1 = n\mathbf{cos}i_2 di_2 \quad (\text{R10})$$

$$\mathbf{cos}i_1' di_1' = n\mathbf{cos}i_2' di_2' \quad (\text{R11})$$

Dividing R10 by R11 yields:

$$\frac{\mathbf{cos}i_1}{\mathbf{cos}i_2} = \frac{\mathbf{cos}i_1'}{\mathbf{cos}i_2'} \quad (\text{R12})$$

Squaring and substituting from R5 and R6 gives:

$$\frac{1-\mathbf{sin}^2i_1}{n^2-\mathbf{sin}^2i_1} = \frac{1-\mathbf{sin}^2i_1'}{n^2-\mathbf{sin}^2i_1'} \quad (\text{R13})$$

which implies

$$i_1 = i_1' \quad \text{and} \quad i_2 = i_2' \quad (\text{R14})$$

Hence, at minimum deviation:

$$i_2 = \frac{\alpha}{2} \quad (\text{R15})$$

$$i_1 = \frac{1}{2}(\delta_{min} + \alpha) \quad (\text{R16})$$

Substituting R15 and R16 into Snell's law at face AB leads to

$$\mathbf{sin} \frac{\delta_{min} + \alpha}{2} = n\mathbf{sin} \frac{\alpha}{2} \quad (\text{R17})$$

from which the refractive index is determined as

$$n = \frac{\mathbf{sin} \frac{\delta_{min} + \alpha}{2}}{\mathbf{sin} \frac{\alpha}{2}} \quad (\text{R18})$$

Specifically, a custom hollow triangular prism with a right-triangle base and an apex angle α of 60° was fabricated and filled with the aqueous ionic liquid solution. Under these geometric conditions, the minimum deviation angle is achieved when the incident angle i_2 equals 30° , corresponding to the scenario in which the emergent ray EF is parallel to the base BC of the triangle. The minimum angle of deflection δ_{min} was measured using a high-precision universal angle protractor. The refractive index at 532 nm (n_{532}) was then calculated using Equation R18.

Since light at 808 nm lies outside the visible spectrum and is unsuitable for the minimum deviation method, the refractive index at 589.3 nm ($n_{589.3}$) was measured using an Abbe refractometer. Subsequently, the refractive index at 808 nm (n_{808}) was estimated by applying the two-wavelength Cauchy dispersion relation to extrapolate from the measured values at 532 nm and 589.3 nm.

$$n = A + \frac{B}{\lambda^2} \quad (\text{R19})$$

Both A and B are constants. All parameters used in these calculations are detailed in Table R1. And the refractive indices at different wavelengths for silicone rubber and nanotape were obtained from the suppliers.

Table R1. Summary of each parameter of refractive index measurement.

	$\alpha/^\circ$	$\delta_{min}/^\circ$	n_{532}	$n_{589.3}$	A	B/nm^2	n_{808}
IL aq	60.00	28.57	1.3964	1.3950	1.3888	2141.6463	1.3921
Silicone rubber	--	--	1.42	--	--	--	1.41
Nanotape	--	--	1.41	--	--	--	1.40

As described in the main text, the transparent elastomer encapsulation surrounding the fiber optic tip can be deformed by varying the volume of injected ionic liquid solution into convex, plain or concave shapes, with curvature precisely tuned to achieve specific divergence angles. Using the refractive index values in Table R1, we performed RSoft simulations of light

propagation at 532 nm for each deformation state of the AOWS head (Figure R4a, b). These simulations demonstrate that the emission angle can be actively modulated by changing the head geometry without compromising overall optical transmission efficiency. We defined the divergence angle as the angle between the emitted beam and the Z axis (positive for divergence, negative for convergence) and found it to vary from -15.13° to $+24.61^\circ$ in simulation, which is fully consistent with the subsequent experimental results (Figure R4c, d).

Figure R4. Divergence angle regulation of AOWS at 532 nm. (a) Simulation of light exiting through five fiber outlets with different curvatures, according to the injection volume. (b) Simulation of power changes during light transmission under various outlet configurations. (c) The optical power and simulation optical transmission efficiency (OTE) under various outlet configurations. (d) The experimental and simulation divergence angles under various outlet configurations. (e) Light shedding area and power density of a parallel incident light exiting AOWS using optical fiber outlet with different features, either convex, flat or concave with

varying curvatures.

To validate these results experimentally, we measured both the divergence angle and the transmitted optical power under different outlet configurations. In the divergence state, the divergence angle was calculated as

$$\mathbf{Divergence\ Angle} = \tan^{-1} \frac{D_1 - D_0}{2d} \quad (\text{R20})$$

and in the convergence state as:

$$\mathbf{Divergence\ Angle} = -\tan^{-1} \frac{D_1 + D_0}{2d} \quad (\text{R21})$$

where D_0 is the diameter of the spot at the fiber head, D_1 is the diameter of the spot on the light screen, and d is the distance from the fiber head to the light screen. All parameters used in these calculations at 532 nm are listed in Table R2. The experimentally measured divergence angles ranged from $-15.38 \pm 0.94^\circ$ to $+25.63 \pm 1.51^\circ$, substantially exceeding the range offered by standard commercial optical fibers. Importantly, the optical output power did not show significant variation across different divergence angles, confirming that beam-angle modulation does not compromise power delivery. As shown in Figure R4e, the concave cap produced a larger illumination area compared with planar or convex configurations. This capability suggests that AOWS can be configured to deliver a desired illumination profile while remaining fixed during interventional phototherapy.

Table R2. Summary of each parameter of divergence angles at 532 nm.

Injection Volume/mL	D_0 /cm	d /cm	$\overline{D_1}$ /cm	S.D./cm	$\overline{Divergence\ Angle}/^\circ$	S.D./ $^\circ$
-0.04			2.22	0.13	25.63	1.51
-0.02			1.00	0.12	9.92	1.69
0.00	0.30	2.00	0.62	0.08	4.572	1.19
0.02			0.46	0.05	-10.76	0.76
0.04			0.80	0.07	-15.38	0.94

We then performed the same set of RSoft simulations and experimental measurements at

808 nm (Figure R5). The overall trends paralleled those observed at 532 nm: both simulated and measured divergence angles varied with head geometry, while optical transmission efficiency remained effectively constant across all configurations. The range of divergence angles at 808 nm was slightly reduced compared with 532 nm, owing to the lower refractive index of the ionic liquid at this wavelength (simulation: -15.37° to $+21.10^\circ$; experiment: $-14.31^\circ \pm 0.60^\circ$ to $+23.99^\circ \pm 1.00^\circ$, Table R3). The precise control of the divergence angle in AOWS enables closer placement of the fiber tip to the target tissue, thereby enhancing negative-feedback regulation and minimizing thermal injury to surrounding healthy tissue.

Figure R5. Divergence angle regulation of AOWS at 808 nm. (a) Simulation of light exiting through five fiber outlets with different curvatures, according to the injection volume. (b) Simulation of power changes during light transmission under various outlet configurations. (c) The optical power and simulation optical transmission efficiency (OTE) under various outlet configurations. (d) The experimental and simulation divergence angles under various outlet

configurations. (e) Light shedding area and power density of a parallel incident light exiting AOWS using optical fiber outlet with different features, either convex, flat or concave with varying curvatures.

Table R3. Summary of each parameter of divergence angles at 808 nm.

Injection Volume/mL	D_0 /cm	d /cm	$\overline{D_1}$ /cm	S.D./cm	$\overline{Divergence\ Angle}/^\circ$	S.D./ $^\circ$
-0.04			2.08	0.08	23.99	1.00
-0.02			0.94	0.11	9.09	1.59
0.00	0.30	2.00	0.54	0.05	3.43	0.78
0.02			0.42	0.08	-10.20	1.16
0.04			0.70	0.04	-14.308	0.60

Q3: The format of the references still needs to be standardized, such as the issue of capitalization in titles, etc.

[Reply to Q3]: Thanks for your kind remind! We have standardized all the format of the references and marked with yellow background in the main text. The revised parts of the references are listed below:

“2. Wei, X., Zhang, C., He, S. *et al.* A dual-locked activatable phototheranostic probe for biomarker-regulated photodynamic and photothermal cancer therapy. *Angew. Chem. Int. Ed.* **61**, e202202966 (2022).

3. Xu, X., Liao, H., Liu, H. *et al.* A water-soluble photothermal host-guest complex with pH-sensitive superlarge redshift absorption. *CCS Chem.* **3**, 2520-2529 (2021).

4. Zhang, L., Dai, Y., Pan, S. *et al.* Copper-selenocysteine quantum dots for NIR-II photothermally enhanced chemodynamic therapy. *ACS Appl. Bio Mater.* **5**, 1794-1803 (2022).

5. Teng, K., Niu, L., Yang, Q. Supramolecular photosensitizer enables oxygen-independent

generation of hydroxyl radicals for photodynamic therapy. *J. Am. Chem. Soc.* **145**, 4081-4087 (2023).

6. Han, P., Xu, H., Zhang, G. *et al.* A processible and ultrahigh-temperature organic photothermal material through spontaneous and quantitative [2+2] cycloaddition-cycloreversion. *Angew. Chem. Int. Ed.* **63**, e202406381 (2024).

7. Zhuang, Z., Qi, Y., Huang, Q. *et al.* Storable polydopamine nanoparticles combined with bacillus calmette-guérin for photothermal-immunotherapy of colorectal cancer. *Adv. Funct. Mater.* **34**, 2404381 (2024).

8. Chu, Y., Liao, S., Liao, H. *et al.* Second near-infrared photothermal therapy with superior penetrability through skin tissues. *CCS Chem.* **4**, 3002-3013 (2022).

9. Gong, F., Cheng, N., Yang, N. *et al.* Bimetallic oxide MnMoOX nanorods for in vivo photoacoustic imaging of GSH and tumor-specific photothermal therapy. *Nano Lett.* **18**, 6037-6044 (2018).

10. Chu, Y., Zhang, W., Yuan, B. *et al.* Deepened photodynamic therapy through skin optical clearing technology in the visible light window. *Langmuir* **40**, 1007-1015 (2024).

11. Meng, Z., Wei, F., Wang, R. *et al.* NIR-laser-switched in vivo smart nanocapsules for synergic photothermal and chemotherapy of tumors. *Adv. Mater.* **28**, 245-253 (2016).

12. Wei, K., Wu, Y., Zheng, X. *et al.* A light-triggered J-aggregation-regulated therapy conversion: from photodynamic/photothermal therapy to long-lasting chemodynamic therapy for effective tumor ablation. *Angew. Chem. Int. Ed.* **63**, e202404395 (2024).

13. Guimarães, C., Ahmed, R., Marques, A. *et al.* Engineering hydrogel-based biomedical photonics: design, fabrication, and applications. *Adv. Mater.* **33**, 2006582 (2021).

14. Chu, Y., Xu, X., Wang, Y. Ultradeep photothermal therapy strategies. *J. Phys. Chem. Lett.* **13**, 9564-9572 (2022).

15. Wu, X., Suo, Y., Shi, H. *et al.* Deep-tissue photothermal therapy using laser illumination at NIR-IIa window. *Nano-Micro Lett.* **12**, 38 (2020).

16. Chu, Y., Wang, Q., Lyu, S. *et al.* Thermal-responsive gel-based overheat limiter enabled intelligent photothermal therapy. *Small* **20**, 2312140 (2024).

20. Choi, M., Humar, M., Kim, S. *et al.* Step-index optical fiber made of biocompatible

- hydrogels. *Adv. Mater.* **27**, 4081-4086 (2015).
22. Fu, R., Luo, W., Nazempour, R. *et al.* Implantable and biodegradable poly(l-lactic acid) fibers for optical neural interfaces. *Adv. Opt. Mater.* **6**, 1700941 (2018).
24. Liu, B., Zhu, H., Zhao, D. *et al.* Hydrogel coating enabling mechanically friendly, step-index, functionalized optical fiber. *Adv. Opt. Mater.* **9**, 2101036 (2021).
25. Zhu, B., Liu, D., Wu, J. *et al.* Slippery core-sheath hydrogel optical fiber built by catalytically triggered interface radical polymerization. *Adv. Funct. Mater.* **34**, 2309795 (2024).
29. Ma, Y., Chu, Y., Lyu, S. *et al.* Injectable optical system for drug delivery, ablation, and sampling in deep tissue. *Adv. Mater. Technol.* **7**, 2101464 (2022).
30. Ran, J., Liu, T., Song, C. *et al.* Rhythm mild-temperature photothermal therapy enhancing immunogenic cell death response in oral squamous cell carcinoma. *Adv. Healthcare Mater.* **12**, 2202360 (2023).
31. Shen, S., Feng, L., Qi, S. *et al.* Reversible thermochromic nanoparticles composed of a eutectic mixture for temperature-controlled photothermal therapy. *Nano Lett.* **20**, 2137-2143 (2020).
32. Xu, H., Huang, G., Cheng, H. *et al.* Thermoelectric-feedback nanocomposite hydrogel for temperature-synchronized monitoring and regulation in accurate photothermal therapy. *Adv. Healthcare Mater.* **13**, 2401609 (2024).
35. Ji, X., Chen, J., Chi, X. *et al.* pH-responsive supramolecular control of polymer thermoresponsive behavior by pillararene-based host – guest interactions. *ACS Macro Lett.* **3**, 110-113 (2014).
36. Chen, L., Zhao, C., Duan, X. *et al.* Finely tuning the lower critical solution temperature of ionogels by regulating the polarity of polymer networks and ionic liquids. *CCS Chem.* **4**, 1386-1396 (2022).
38. Wu, S., Zhang, Q., Deng, Y. *et al.* Assembly pattern of supramolecular hydrogel induced by lower critical solution temperature behavior of low-molecular-weight gelator. *J. Am. Chem. Soc.* **142**, 448-455 (2020).
39. Zou, Y., Brooks, D., Kizhakkedathu J. A novel functional polymer with tunable LCST. *Macromolecules* **41**, 5393-5405 (2008).

40. Dan, M., Su, Y., Xiao, X. *et al.* A new family of thermo-responsive polymers based on poly[N-(4-vinylbenzyl)-N, N-dialkylamine]. *Macromolecules* **46**, 3137 – 3146 (2013).
41. Zhu, Y., Batchelor, R., Lowe, A. *et al.* Design of thermoresponsive polymers with aqueous LCST, UCST, or both: modification of a reactive poly(2-vinyl-4,4-dimethylazlactone) scaffold. *Macromolecules* **49**, 67-680 (2016).”

Reviewer #2: I co-reviewed this manuscript with one of the reviewers who provided the listed reports. This is part of the Nature Communications initiative to facilitate training in peer review and to provide appropriate recognition for Early Career Researchers who co-review manuscripts.

[Reply to Reviewer #2] Thank you very much for your time and efforts in reviewing this manuscript, your suggestions and comments give us inspirations to further improve the work.

Reviewer #3: Phototherapy, valued for its non-toxicity, selectivity, and minimal trauma, is predominantly applied to treat superficial diseases due to the limited penetration of light through tissues. While optical-fiber-assisted interventional phototherapy addresses this limitation, it lacks an immediate mechanism to mitigate overheating of surrounding healthy tissues. This manuscript introduces an adaptive optical waveguide system (AOWS) based on a thermoresponsive ionic liquid with a tunable lower critical solution temperature (LCST), which facilitates deep-tissue phototherapy while mitigating overheating of surrounding healthy tissues through a negative feedback mechanism. The system addresses critical challenges in conventional phototherapy, such as extended light paths and inadvertent thermal damage to non-target tissues, thereby offering a safer and more efficient solution for deep-tissue applications. The data and characterization presented robustly support the conceptual framework and system design of the authors. Given the novelty and comprehensive characterizations of the proposed system, I recommend its publication in Nature Communications following minor revisions.

[Reply to Reviewer #3] Thank you very much for your positive comments and detailed modification suggestions, which helps a lot on the improvement of this manuscript. According to your suggestions, we have supplemented several necessary experiments and revised the manuscript carefully. Our point-to-point responses to all your concerns are as below:

Q1. The authors assert that the fiber employed in the system protects biological tissues by thermally safeguarding their structure; however, biological tissues are also sensitive to external mechanical stresses, such as friction. It is unclear whether the fiber tip is rigid or flexible, and it would be valuable to include tests of the mechanical properties of the fiber tip, such as its modulus.

[Reply to Q1]: Thank you for your insightful consideration. In response to your suggestions, we have augmented our study with a comprehensive mechanical characterization of the optical fiber tip. The tip assembly comprises a silicone-rubber sleeve, a nanotape layer, and an internal aqueous ionic liquid solution. Given the practical constraints of testing and the predominant influence of the silicone rubber sleeve on overall mechanics, we focused our detailed analysis

on this component. Dynamic mechanical analysis (DMA) yielded tensile, compressive, and flexural modulus of 6.23 MPa, 6.51 MPa, and 10.32 MPa, respectively (Figure R6a-c). Furthermore, we compared the Young's modulus of our fiber tip with those of alternative materials commonly used in fiber fabrication (Figure R6d)^[1-15]. The Young's modulus of the tip is several orders of magnitude lower than that of commercially available silicon-based and polymer fibers (e.g. PLA and PMMA), and closely matches that of elastomers and certain hydrogels. This modulus compatibility reduces mechanical mismatch with biological tissues and minimizes implantation-related damage arising from material rigidity.

Figure R6. Mechanical properties of the optical fiber tip. (a) Tensile modulus. (b) Compression modulus. (c) Bending modulus. (d) Young's modulus of the optical fiber tip relative to other already published materials^[1-15].

Q2. The paper describes the development of an adaptive optical waveguide system (AOWS) for deep phototherapy with tissue protection, leveraging a negative feedback modulation mechanism. While this concept is intriguing, the authors do not clearly define the mechanism or compare its advantages to conventional thermal protection strategies. A more detailed discussion of the concept, along with additional evidence supporting its benefits, would strengthen the manuscript and clarify this novel approach.

[Reply to Q2] Thank you for highlighting the need to more clearly define our negative feedback modulation mechanism and to position it relative to existing thermal-protection strategies. We have supplemented the following discussion to clearly define the mechanism and compare its advantages to conventional thermal protection strategies.

Negative feedback regulation refers to the process by which a portion of a system's output is returned to the input in an inverted form; when the output deviates from the target, the feedback signal suppresses the deviation, driving the system back toward equilibrium. It plays a critical role in maintaining homeostasis, enhancing robustness, and improving disturbance rejection (*Nat. Nanotechnol.* **2022**, 17, 1303–1310). It is precisely because of these advantages that we have incorporated this mechanism into the thermal protection system to enhance the safety of interventional photothermal therapy. The following section will provide a detailed comparison of the advantages of our method compared to conventional thermal protection strategies.

In recent years, photothermal therapy (PTT) has progressed toward precise temperature monitoring and autonomous thermal control. For accurate temperature monitoring, Wei et al. adjusted irradiation power density and exposure duration, employing repeated irradiation cycles to maintain therapeutic efficacy while avoiding thermal injury (*Adv. Healthcare Mater.* **2023**, 12, 2202360). Zheng et al. developed a photothermoelectric cobalt-infused chitosan nanocomposite hydrogel that exhibits a linear correlation between temperature shifts and resistance changes, enabling precise subcutaneous temperature measurements to prevent overheating (*Adv. Healthcare Mater.* **2024**, 13, 2401609). However, these approaches cannot respond instantaneously to thermal excursions and instead rely on manual adjustments of laser power based on real time temperature feedback.

In the area of autonomous temperature regulation, Zhu et al. proposed a photomedicine strategy using a temperature adaptive hydrogel optical fiber waveguide (*Nat. Commun.* **2022**, 13, 7789). This strategy harnesses the phase transition of a responsive material as an optical switch to limit the core temperature of the photothermal agent. Because the fiber tip and the photothermal medium are in intimate contact, thermal equilibrium is reached rapidly. In practice, however, variations in tissue thermal conductivity and laser power produce substantial

differences in the resulting steady state thermal distribution and ablation volume. Ensuring that the treated region precisely matches the lesion geometry therefore requires extremely fine tuning of both the phase transition temperature and the laser output. Without such precision, neighboring healthy tissues remain vulnerable to unintended thermal damage, a challenge that is difficult to surmount in clinical applications.

In our design, the photothermal material is separated by a small gap from the fiber tip, which is loaded with the phase-change material, rendering heat transfer the rate-limiting step. So the negative feedback mechanism goes like this: when the tip temperature exceeds the phase - change threshold, scattering within the phase - change medium blocks the optical path, arresting further heating and safeguarding healthy tissue; conversely, if the tip temperature remains below this threshold, optical transmission continues unabated, allowing the lesion temperature to rise and induce thermal coagulative necrosis. Because this mechanism relies on a dynamically evolving thermal field rather than a fixed steady-state, its effectiveness is essentially independent of laser power. It requires only that the phase-change temperature be set between the tolerance limits of normal and target tissues and positioned at the edge of the intended ablation zone. By limiting the thermal kill region to these bounds, our approach greatly reduces the risk of unintended thermal injury to surrounding healthy tissue.

Overall, through its feedback-based regulation mechanism, AOWS can provide thermal protection that operates almost like **an "on-off" switch, without relying on irradiation power.**

Thank you again for your constructive suggestion. The relevant content has been revised carefully in the main text and marked with yellow background.

Q3. The "simulated tissue" used in the study (cross-linked gelatin with polypyrrole nanoparticles) lacks explicit synthesis parameters, such as the size distribution of the nanoparticles. Furthermore, the photothermal conversion efficiency of the photothermal material is a critical parameter that should be measured and reported, as it plays a vital role in evaluating the performance of the system in thermal protection for photothermal therapy.

[Reply to Q3]: Thank you for your careful consideration. We fully agree that detailed synthesis parameters for both the simulated tissue and the polypyrrole nanoparticles (PPy NPs) are

essential, and have accordingly included a comprehensive description of their preparation.

The supplemented part in Methods section:

“To fabricate the simulated tissue model, 5.0 mL of a 10 wt.% gelatin solution was dispensed into a Petri dish (3.50 cm diameter) and crosslinked by addition of 100 μ L of 10 wt.% glutaraldehyde. Crosslinking was carried out at 4 $^{\circ}$ C for 24 h to ensure uniform network formation. Thereafter, a central cavity (\varnothing 0.80 cm) was created using a biopsy punch, and 0.40 mL of 10 wt.% gelatin solution containing 1.0 mg/mL polypyrrole was introduced into the cavity. The composite construct was then subjected to the same 4 $^{\circ}$ C, 24 h glutaraldehyde crosslinking protocol to yield the final simulated tissue.”

“PPy NPs were synthesized by the emulsion method. 10.0 mL of water containing 0.5 wt.% polyvinyl alcohol (PVA 1788) was used as the aqueous phase, and the oil phase was 2.0 mL of methylene chloride (DCM) dissolving 100.0 μ L of pyrrole. Oil-in-water emulsions were produced by a high-speed disperser, followed by adding 0.40 mL of KPS aqueous solution (2.0 M) to initiate the polymerization of pyrrole in the oil phase. Then the DCM was removed by volatilization. Finally, the polypyrrole nanoparticles (PPy NPs) dispersion was obtained after a dialysis purification.”

The particle-size distribution of the PPy NPs was characterized by dynamic light scattering (Figure R7a). The average hydrodynamic diameter was 127.8 nm with a polydispersity index (PDI) of 0.100, confirming the high uniformity of the emulsion-synthesized nanoparticles.

The photothermal conversion efficiency (η_{PT}) of the PPy NPs was determined following our previously reported methodology (*Chem. Eur. J.* **2016**, 22, 1152). First, the UV-Vis-NIR absorption spectrum of a 0.08 mg/mL PPy NPs suspension was recorded to obtain the absorbance at 808 nm (Figure R7b). Next, 1.0 g of this suspension was placed in a quartz cuvette and irradiated at five different power densities. Each irradiation lasted two minutes after the sample had equilibrated to its initial temperature, and five consecutive heating-cooling cycles were performed at each power setting to evaluate reproducibility (Figure R7c-d). The typical η_{PT} at a fixed power was calculated according to Equation. R22:

$$\eta_{PT} = \frac{hA\Delta T_{max}}{I(1-10^{-A\lambda})} \quad (R22)$$

Where h is the heat transfer coefficient, A is the surface area of the system, ΔT_{max} is the

temperature difference between the maximum temperature and initial temperature, I is the light power, A_λ is the absorbance at wavelength λ . In this equation, only hA is unknown. In order to get hA , θ defined as the ratio of ΔT to ΔT_{max} was introduced and the value of hA is derived according to equation (R23):

$$t = -\frac{\sum_i m_i C_{p,i}}{hA} \ln \theta \quad (\text{R23})$$

Therefore, hA can be determined by the linear time data from the cooling period t vs $-\ln \theta$. The weight (m) of PPy NPs solution was 1.0 g and specific heat (C_p) was determined to be 4.20 J/(g \cdot $^\circ\text{C}$). Substituting hA value into Equation. R1, the photothermal conversion efficiency (η_{PT}) could be calculated.

Figure R7. Characterization of polypyrrole nanoparticles (PPy NPs). (a) Particle size of PPy NPs, inset is a photograph of PPy NPs particle dispersion. (b) Absorption spectra of PPy NPs. Temperature rise of 0.08 mg/mL solution of PPy NPs under 808 nm laser irradiation with

different powers (c) and infrared thermal imaging photographs (d).

We selected the second heating-cooling cycle at each irradiation power to calculate the corresponding photothermal conversion efficiency ($\eta_{PT,I}$). All parameters used in these calculations are detailed in Table R4. The mean photothermal conversion efficiency of the PPy NPs was determined to be $42.27\% \pm 2.92\%$, which agrees well with literature values (*Chem. Commun.*, **2012**, 48, 8934-8936). Notably, polypyrrole is a biocompatible photothermal agent that, unlike many metal-based nanoparticles such as gold, exhibits excellent cyclability and resists photobleaching (*Adv. Mater.* **2013**, 25: 777-782). These attributes ensure the reliability and reproducibility of our thermal protection experiments.

Table R4. Summary of each parameter of photothermal conversion efficiency.

I/W	A_{808}	$\Delta T_{\max}/^{\circ}\text{C}$	$\frac{\sum_i m_i C_{p,i}}{hA}/\text{s}$	$\sum_i m_i C_{p,i}/\text{J}\cdot^{\circ}\text{C}^{-1}$	$hA/\text{W}\cdot^{\circ}\text{C}^{-1}$	$\eta_{PT,I}$	$\overline{\eta_{PT}}$	S.D.
0.2		2.1	131.9953		0.0318	40.98%		
0.4		4.1	123.4600		0.0340	42.77%		
0.6	0.7334	6.6	135.4013	4.20	0.0310	41.85%	42.27%	2.92%
0.8		8.2	112.7826		0.0372	46.82%		
1.0		10.0	132.3584		0.0317	38.92%		

References:

- [1] Toland, K., Conway, A., Cunningham, L. *et al.* Development of a pulling machine to produce micron diameter fused silica fibres for use in prototype advanced gravitational wave detectors. *Class. Quantum Grav.* **35**, 165004 (2018).
- [2] Ilyas, R., Zuhri, M., Aisyah, H. *et al.* Natural fiber reinforced polylactic acid, polylactic acid blends and their composites for advanced applications. *Polymers* **14**, 202 (2022).
- [3] Yang, D., Yu, J., Tao, X. *et al.* Structural and mechanical properties of polymeric optical fiber. *Mater. Sci. Eng. A* **364**, 256–259 (2004).
- [4] Wang, M., Jin, H., Kaplan, D. *et al.* Mechanical properties of electrospun silk fibers. *Macromolecules* **37**, 6856-6864 (2004).
- [5] Agnarsson, I., Kuntner, M., Blackledge, T. Bioprospecting finds the toughest biological material: extraordinary silk from a giant riverine orb spider. *PLoS One* **5**, e11234 (2010).
- [6] Shan, D., Zhang, C., Kalaba, S. *et al.* Flexible biodegradable citrate-based polymeric step-index optical fiber. *Biomaterials* **143**, 142-148 (2017).
- [7] Guo, J., Niu, M., Yang, C. Highly flexible and stretchable optical strain sensing for human motion detection. *Optica* **4**, 1285-1288 (2017).
- [8] Feng, J., Zheng, Y., Bhusari, S. *et al.* Printed degradable optical waveguides for guiding light into tissue. *Adv. Funct. Mater.* **30**, 2004327 (2020).
- [9] Feig, V., Tran, H., Lee, M. *et al.* Mechanically tunable conductive interpenetrating network hydrogels that mimic the elastic moduli of biological tissue. *Nat. Commun.* **9**, 2740 (2018). .
- [10] Zhu, B., Liu, D., Wu, J. *et al.* Slippery core-sheath hydrogel optical fiber built by catalytically triggered interface radical polymerization. *Adv. Funct. Mater.* **34**, 2309795 (2024).
- [11] Guo, J., Liu, X., Jiang, N. *et al.* Highly stretchable, strain sensing hydrogel optical fibers. *Adv. Mater.* **28**, 10244–10249 (2016).
- [12] Liu, B., Zhu, H., Zhao, D. *et al.* Hydrogel coating enabling mechanically friendly, step-index, functionalized optical fiber. *Adv. Opt. Mater.* **9**, 2101036 (2021).
- [13] Zhang, J., Zhou, J. The detection and evaluation of vascular stiffness. *Acta Physiologica Sinica* **74**, 894–902 (2022).
- [14] Radulescu, D., Peride, I., Petcu, L. *et al.* Supersonic shear wave ultrasonography for assessing tissue stiffness in native kidney. *Ultrasound Med. Biol.* **44**, 2556-2568 (2018).

[15] Calhoun, M., Bentil, S., Elliott, E. *et al.* Beyond linear elastic modulus: viscoelastic models for brain and brain mimetic hydrogels. *ACS Biomater. Sci. Eng.* **5**, 3964–3973 (2019).

Replies to Reviewers' comments and questions

Reviewer #3: The current version after revision could be accepted as is.

[Reply to Reviewer #3] Thank you very much for your time and efforts in reviewing this manuscript.

Reviewer #4: The manuscript presents an adaptive optical waveguide system (AOWS) that employs an ionic-liquid aqueous solution with a tunable lower critical solution temperature (LCST). Above a preset temperature the solution undergoes a reversible, rapid phase transition to an opaque state, producing a temperature-triggered negative-feedback gating mechanism that scatters transmitted light. This mechanism mitigates overheating of surrounding healthy tissue while preserving effective photothermal treatment at the target site. Unlike conventional passive thermal management or externally driven modulation, the proposed strategy implements an autonomous, local safety control that is conceptually novel and practically deployable for photothermal therapy. Experimentally, the system achieves near ‘on–off’ thermal protection that is independent of irradiation power. Moreover, a tailored fiber-optic tip permits controlled adjustment of beam divergence, enabling large-area subcutaneous illumination with tunable treatment radii. In addition, comprehensive material characterization, quantifiable temperature-gating behavior, and the divergence-angle design of the manuscript are corroborated by thermal and optical simulations as well as ex-/in-vivo demonstrations, which together substantiate the mechanism and therapeutic efficacy. High-quality figures and a rigorous, quantitative presentation further strengthen the credibility of the proposed mechanism and the conclusions. Therefore, I strongly recommend its publication in Nature Communications.

[Reply to Reviewer #4] Thank you very much for your time and efforts in reviewing this manuscript.